# Optimizing multi-user indoor sound communications with acoustic reconfigurable metasurfaces

Hongkuan Zhang [1,5], Qiyuan Wang [1,4,5], Mathias Fink [2] &
Guancong Ma [1,3] ✉

Sound in indoor spaces forms a complex wavefield due to multiple scattering encountered by the sound. Indoor acoustic communication involving multiple sources and receivers thus inevitably suffers from cross-talks. Here, we demonstrate the isolation of acoustic communication channels in a room by wavefield shaping using acoustic reconfigurable metasurfaces (ARMs) controlled by optimization protocols based on communication theories. The ARMs have 200 electrically switchable units, each selectively offering 0 or $\pi$ phase shifts in the reflected waves. The sound field is reshaped for maximal Shannon capacity and minimal cross-talk simultaneously. We demonstrate diverse acoustic functionalities over a spectrum much larger than the coherence bandwidth of the room, including multi-channel, multi-spectral channel isolations, and frequency-multiplexed acoustic communication. Our work shows that wavefield shaping in complex media can offer new strategies for future acoustic engineering.

Most indoor spaces are complex acoustic cavities, wherein the sound fields are scrambled by reflections and multiple scattering[1]. Such environments are never ideal for acoustic communication: the multiple scattering leads to cross-talk, the disorder garbles conversations and decreases speech intelligibility. In this work, by a successful crossover of adaptive wavefield shaping[2,3], acoustic metasurfaces[4,5], information and communication theories[6,7], we experimentally demonstrate the control of complex indoor sound fields for optimal acoustic communications between multiple sources and receivers by acoustic reconfigurable metasurfaces (ARMs) that provides binary phase control. The idea is to modify the room environment by wavefield shaping to physically optimize multiple communication channels between various sources and receivers.

Up to now, adaptive wavefield shaping has revolutionized the control of light[8–10], microwave[11–13], and sound[14,15] in complex media. A plethora of functionalities have been realized, such as focusing and imaging through opaque materials[3,16,17], perfect transmission through disordered media[18,19], depth-targeted energy delivery[20], spatiotemporal control of complex fields[11–14], chaos-assisted analog computing[21,22]. In particular, adaptive wavefield shaping can either synthesize the input wavefield or modify the complex media such that an input wave optimally couples to open transmission eigenchannels of a medium for high transmission efficiency[18,19]. Such an approach has been shown to benefit microwave-based communications[13]. However, unlike telecommunications that can benefit from signal processing provided by sophisticated modern electronics, such as filtering, sound communications are directly conducted among humans who do not naturally process such capabilities. The phenomenon of the cocktail party effect in the human auditory system allows individuals to selectively attend to specific sounds[23], facilitating the reduction of cross-talk in multichannel communications. Nevertheless, in complex environments, the cognitive capacity of the human perception system is limited. For this reason, optimal sound communications present a unique set of challenges and are so far beyond reach in complex environments. Here, to optimize sound communications and information transfer between multiple sources and receivers in a room, we first measure the

[1]Department of Physics, Hong Kong Baptist University, Kowloon Tong, Hong Kong, China. [2]Institut Langevin, ESPCI Paris, Université PSL, CNRS, Paris, France. [3]Shenzhen Institute for Research and Continuing Education, Hong Kong Baptist University, Shenzhen 518000, China. [4]Present address: Graduate School of Engineering, The University of Tokyo, Tokyo, Japan. [5]These authors contributed equally: Hongkuan Zhang, Qiyuan Wang. ✉e-mail: phgcma@hkbu.edu.hk

multi-spectral channel matrix that connects, at each frequency, the sources and receivers in the room. It encapsulates the disordered nature of the complex sound field but contains few degrees of freedom, and therefore it is easy to handle. We measure this channel matrix for different configurations of ARMs. The first of its kind, the ARMs modify the reflection phase and function as tunable mirror that on-demand control the phases of the reflected waves, which effectively alter boundary conditions of a portion of the room. Each unit cell of the metasurfaces provides, on demand, a two-state phase shift (0 or π). By driving the ARMs using optimization schemes that target selected properties of the channel matrices, we successfully demonstrate diverse functionalities, including channel isolation and cross-talk elimination, frequency-multiplexed channel conditioning, as well as other, more flexible controls for acoustic communications. In particular, we demonstrate the control of multi-spectral sound fields covering a spectrum much larger than the coherence bandwidth of the room and the striking effect of crosstalk-free simultaneous music playback with two sources, each playing a different music piece. Our work opens broad horizons for future sound-scaping and other acoustic engineering.

## Results

### Channel matrix and conditions of optical channel isolation

**Channel matrix for acoustic communication in a complex acoustic environment.** For each sound frequency $f$, a channel matrix, denoted $\mathbf{H}(f)$ with $h_{ij}$ as entries, directly connects the sources and receivers by $\mathbf{R} = \mathbf{H} \cdot \mathbf{S}$, where $\mathbf{S}$ and $\mathbf{R}$ are the source and receiver vectors. A simple example is shown in Fig. 1a, which has two loudspeakers (sources) and two microphones (receivers). Apparently, both $\mathbf{S}$ and $\mathbf{R}$ are $2 \times 1$ vectors, and the channel matrix is $2 \times 2$ in dimension[6]. In general, the sounds picked up by the two receivers are mixtures of signals emitted from the two sources that are further garbled by the multiple scatterings by the boundaries and various objects. Our lab is a furnished room with an irregular shape (Fig. 2a). It is a random media in the reverberating regime, and the sound field inside is disordered in character (see Methods and Supplementary Note 2 for more details). Therefore, $h_{ij}$ are suitably represented by complex random numbers[6,24]. Note that $\mathbf{H}$ generically has no symmetry and is not Hermitian. Thus, its eigenvectors, if exist, are not orthogonal in general, i.e., the eigenchannels are generically not independent. It follows that one cannot achieve channel separation by choosing $\mathbf{S}$. Instead, we must alter $\mathbf{H}$ itself.

According to Shannon's law in information theory, the optimal channel capacity is determined by the singular value distribution of the channel matrix[6]. Consider an $N \times N$ channel matrix, to achieve maximum channel capacity, i.e., $N$ independent channels, the channel matrix is required to have maximum entropy, which is directly related to the effective rank of $\mathbf{H}$ as[25]

$$R_{\mathrm{eff}}(\mathbf{H}) = \exp(E), \tag{1}$$

where $E = -\sum_{k=1}^{N} p_k \ln p_k$ is the Shannon entropy, and $p_k = \sigma_k / (\sum_{i=1}^{N} \sigma_i)$ are the normalized singular values of $\mathbf{H}$. A higher effective rank indicates a greater number of independent eigenchannels available in the channel matrix. For an $N \times N$ channel matrix, the effective rank theoretically ranges from 1 to $N$. Upon reaching the full effective rank, all $p_k$ become identical, indicating that the eigenvectors of $\mathbf{H}$ are nearly orthogonal. In this case, the channels are minimally mixed. Therefore, the first goal of our approach is to achieve channel independence by maximizing $R_{\mathrm{eff}}$ for a given acoustic configuration.

However, channel independence alone is insufficient for acoustic communications because, even with independent channels, the receivers can still concurrently detect signals from multiple sources. This does not pose a problem for telecommunication scenarios because once $\mathbf{H}$ is known, such mixing (wave superposition) can be

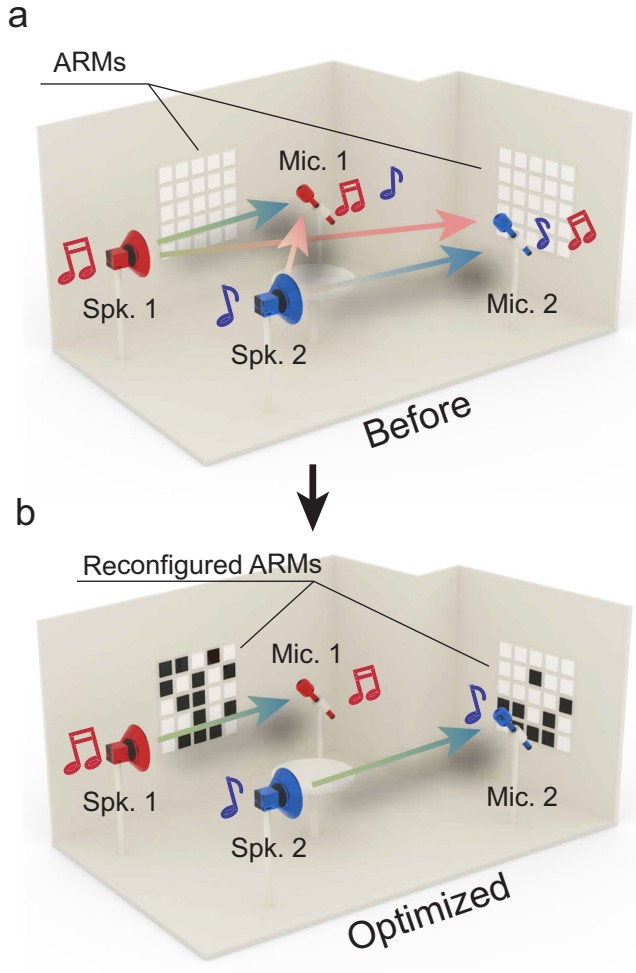

**Fig. 1 | Channel conditioning for acoustic communications. a** Acoustic channels in a room are generically coupled, so each microphone (Mic.) captures the sound from both loudspeakers (Spk.). **b** Independent, isolated channels can be achieved by wavefield shaping using the acoustic reconfigurable metasurfaces (ARMs), such that loudspeakers and microphones communicate without interference from others.

removed by either tailoring the emission or by signal post-processing. However, because acoustic communications commonly involve humans, who obviously lack such signal-processing capabilities, the acoustic channels have to be further optimized to eliminate signal superpositions. For example, in Fig. 1b, it is ideal for microphone 1 to only detects the acoustic signal from loudspeaker 1 but nothing from loudspeaker 2. This requires $\mathbf{H}$ to take a diagonal form. Hence, we introduce a second parameter $w_1$ that characterizes the degree of diagonalization

$$w_1(\mathbf{H}) = \frac{\sum_{i \neq j} |h_{ij}|}{\sum_{i = j} |h_{ij}|}. \tag{2}$$

Obviously, when Eq. (2) vanishes, $\mathbf{H}$ is diagonal.

The two considerations together give an objective function

$$\mathcal{G}_1(\mathbf{H}) = [N - R_{\mathrm{eff}}(\mathbf{H})] + w_1. \tag{3}$$

The minimization of $\mathcal{G}_1$ should yield a system that not only reaches maximal channel capacity but also produces channels that offer

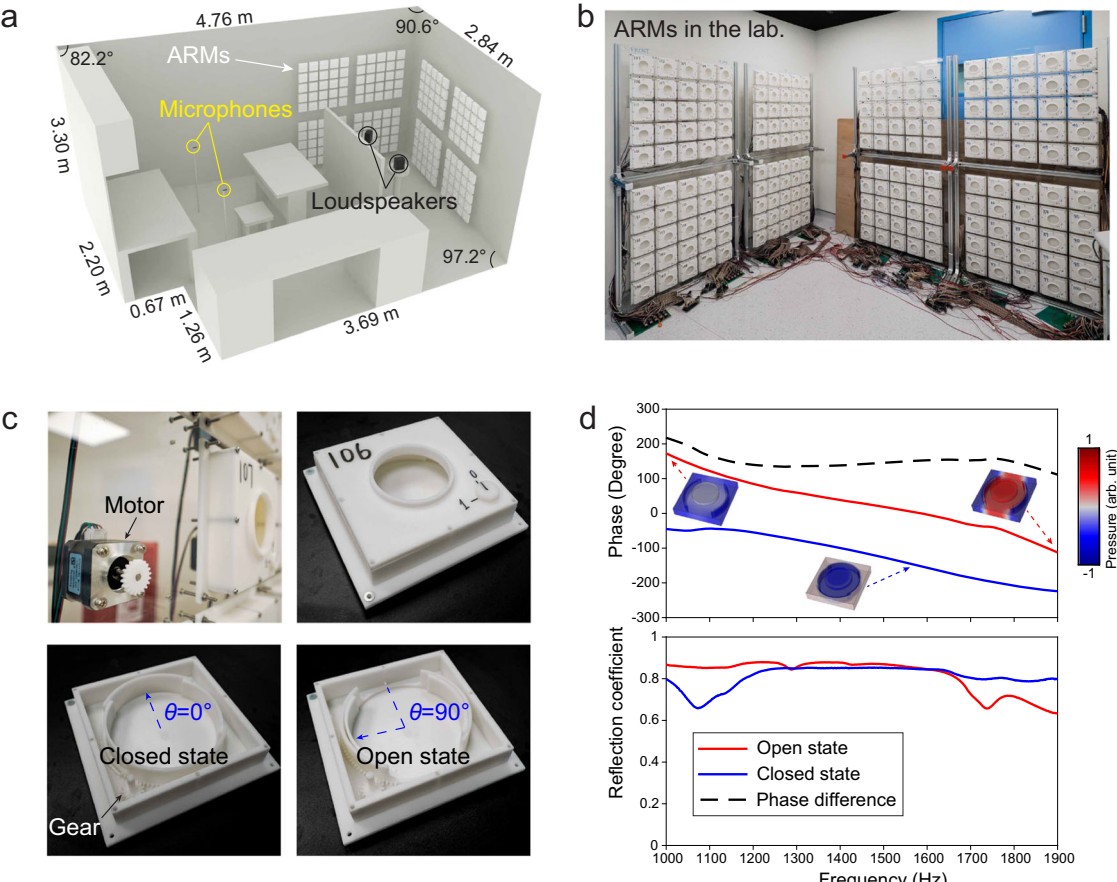

**Fig. 2 | The experimental environment and the design of the ARMs. a** The experimental environment, which is a furnished room of an irregular shape in the reverberating regime. **b** A photo of the ARMs consisting of a total of 200 tunable Helmholtz resonators. **c** Photos of the THRs that form the ARMs. The volume of the THRs can be altered by rotating the internal partition with a program-controlled motor. The upper panels show the motor mounted on a reflective backplate (transparent) and the external view of the THR. The lower panels show the closed and open states. **d** Experimentally measured reflection phases and amplitude reflection coefficients of the THR at closed (blue) and open (red) states. The black dashed curve plots the phase difference between 2 states. The insets show the mode profiles obtained using finite-element simulation.

one-to-one signal delivery between sources and receivers. We denote this condition as optimal channel isolation (OCI).

We remark that the minimization of either $R_{eff}$ or $w_1$ alone is insufficient for achieving OCI, and it is necessary to minimize both of them simultaneously. For example, minimizing $w_1$ alone, i.e., without enforcing a maximum $R_{eff}$, can still reduce off-diagonal entries. But there is no guarantee that the diagonal entries have near-equal values. If the diagonal entries differ significantly, the two channels have drastically different signal-to-noise ratios, which is not optimal for communication purposes. For a detailed discussion on this issue and additional experiments, please refer to Supplementary Note 4.

**Achieving OCI via adaptive wavefield shaping**
Because the channel matrix encompasses disordered characteristics of the complex sound field and the multipath transmission of acoustic signals, the only way to control it is to alter the environment. Our previous works have already demonstrated such possibilities by extending wavefront shaping—a powerful technique previously used for controlling the propagation of light in multiple-scattering propagation—for airborne sound[14,15]. Here, we develop a set of ARMs to serve as the sound-modulating device. The ARMs are based on tunable acoustic metasurfaces and are integrated as a part of the boundaries of the room (Fig. 2b). They consist of 200 units of tunable Helmholtz resonators (THRs)[4,5,26], each with independently tunable resonance. The design of the THRs is shown in Fig. 2c. Simply put, the volume of

the THR is actively adjustable by an electric motor, which shifts its resonant frequency between two values. As a result, the reflection phase can be actively tuned between 0° and ~160° over a broad frequency range of 1100–1850 Hz, which exceeds 3/4 octave, as shown in Fig. 2d. This change in reflection phases alters the waves that form the disordered sound field in the room, by which the channel matrix optimization is performed. See Methods for more details on the design of the ARMs.

Loudspeakers and microphones play the roles of sources and receivers. The channel matrix is determined by experimentally measuring the transfer functions between each loudspeaker and microphone. The spatial separations among the loudspeakers, and among the microphones, are larger than the correlation length, which is about half the wavelength. The distance between any loudspeaker and microphone is larger than the reverberating radius so that direct sound does not dominate the transfer functions (see Methods for more details). First, as a proof of principle, we demonstrate the 2-channel OCI of single-frequency sound at 1300 Hz. This is achieved by performing the ARMs using a climbing algorithm targeting the minimization of $\mathcal{G}_1(\mathbf{H})$. The results of 40 independent realizations with uncorrelated configurations are summarized in Fig. 3. Figure 3a, b compare $|h_{ij}|$ before and after the wavefield shaping. It is clearly seen that the evenly distributed entries are put to the diagonal and fall on the unit circle on the complex plane, and off-diagonal terms are suppressed to near zero. In Fig. 3c, we see that the process indeed raises

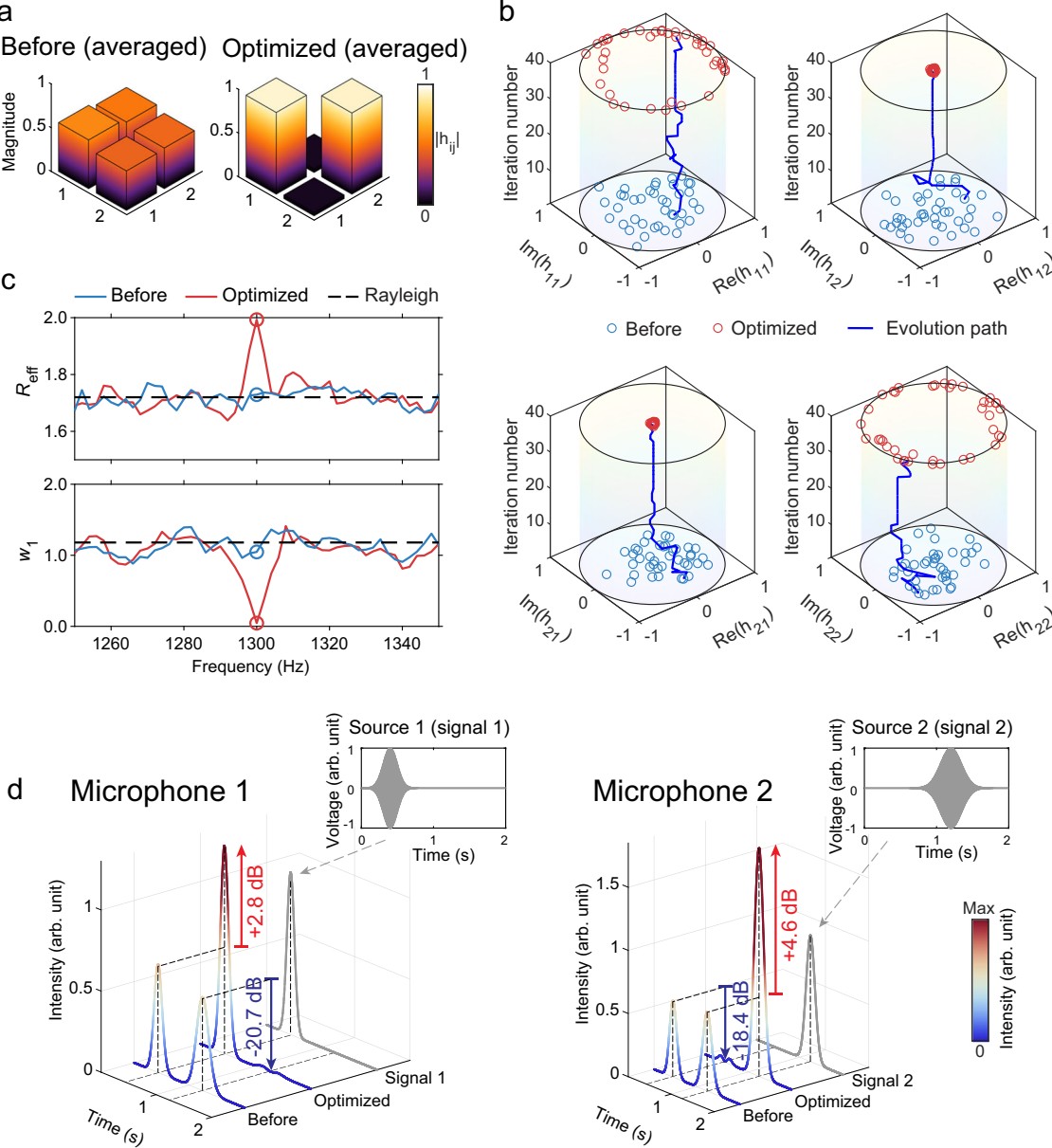

**Fig. 3 | Single-frequency two-channel OCI. a** The averaged entries of channel matrix before and after achieving OCI. **b** The optimization processes drive the diagonal entries of the channel matrices to the unit circle on the complex plane, and the off-diagonal entries to zero. **c** Upon attaining OCI, notable changes in the effective ranks $R_{eff}$ and the degree of diagonal $w_1$ are observed near 1300 Hz. The black dashed lines show the prediction based on Rayleigh channels. **d** The auditory effect of OCI. The two insets show the two temporally separated beeps are emitted by two loudspeakers. The envelops of the signal received by microphone 1 (left) and microphone 2 (right). It is clear that prior to OCI, both microphones detect the sound from both sources (double peak). After OCI, microphones 1 and 2 to receive the sound from the corresponding loudspeaker, and cross-talks are considerably suppressed (single peak). The gray curves depict the emission.

$R_{eff}$ to the theoretical upper bound of 2, and in the meantime, $w_1$ vanishes. These values are significantly different from their typical values, which, on average, converge to the prediction of Rayleigh channels[6] [see Supplementary Note 5]. The bandwidth of the optimization effect is roughly ±4 Hz, which is consistent with the coherence bandwidth of the room. It is essential to point out that optimizing the channel isolation metric at a single frequency does not statistically affect the channel metrics at other frequencies beyond the coherence bandwidth. Please refer to Supplementary Note 6 for details. To demonstrate the effect of the OCI, we send with two loudspeakers two temporally separated "beeps" (finite-duration, gaussian-enveloped trains of sine waves centered at 1300 Hz) and record the signals detected by two microphones. The results are plotted in Fig. 3d. Prior to optimization, both microphones receive two beeps, which is well

expected. After OCI is obtained, both microphones only detect one beep and microphone 1(2) only picks up the beep from loudspeaker 1(2). The intensities of the desired signals received by microphones 1 and 2 have increased by 2.8 dB and 4.6 dB, respectively, and the intensities of the unwanted signals are significantly suppressed by 20.7 dB and 18.4 dB, respectively. We remark that the OCI effect does not depend on the forms of acoustic signal from the sources, i.e., it makes no difference if continuous sound or temporally overlapped pulses are used instead. The purpose of using temporally separated signals is for better visual comparison in the figures.

To further show the effectiveness of our approach, we compared the energy delivered by the channels before and after OCI, which can be easily extracted from the entries of the channel matrices. When OCI is attained, the energy delivered by the intended channels is enhanced

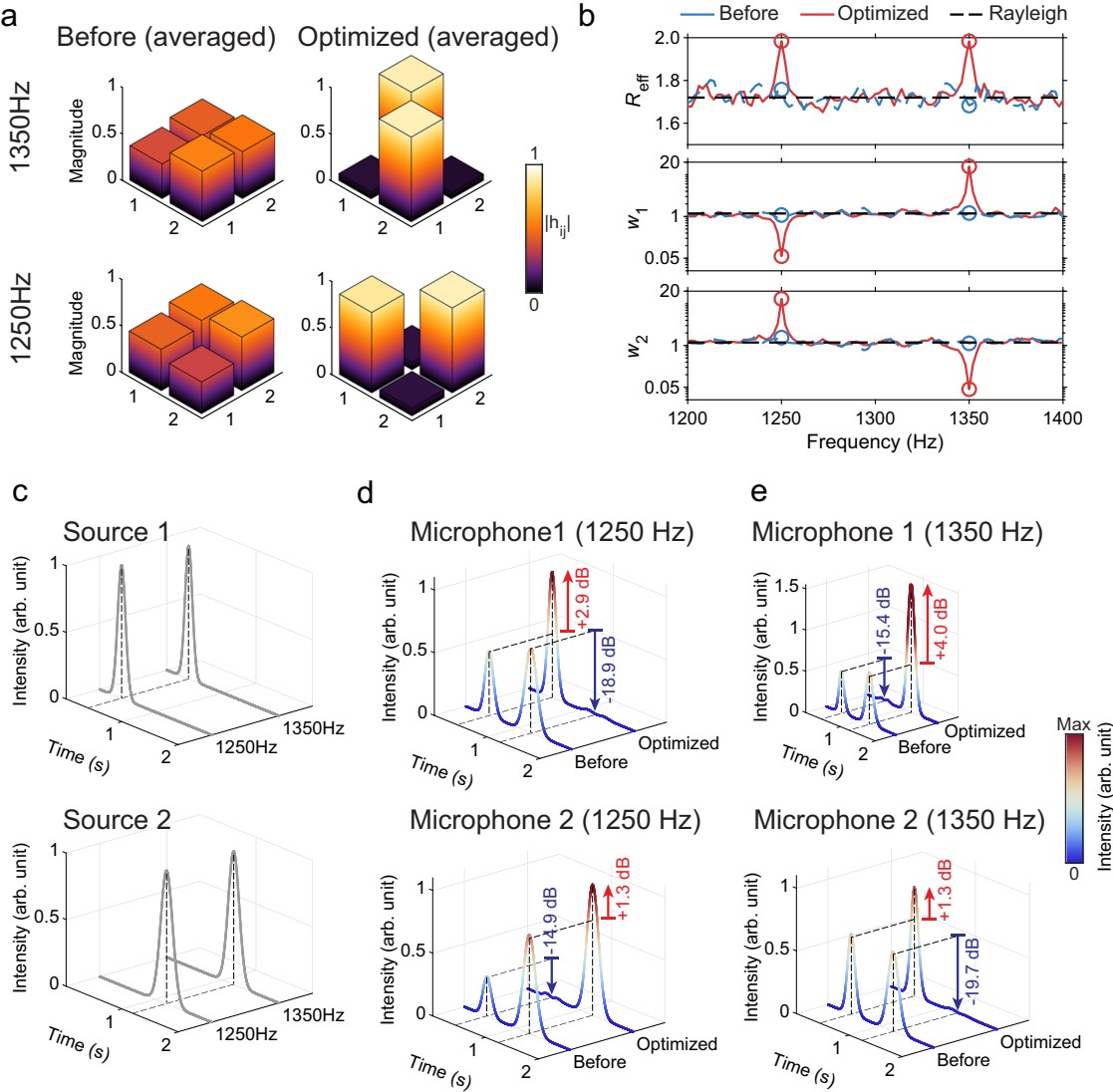

**Fig. 4 | Dual-frequency OCI. a** The diagonal (off-diagonal) entries of the channel matrix are maximized at 1250 (1350) Hz. **b** The effective ranks and other objective parameters ($w_1$ and $w_2$) prior and after OCI. Here, $w_1$ and $w_2$ draw a logarithmic scale. **c** The signal intensities emitted by loudspeakers 1 and 2. **d, e** The averaged

signals received by microphones 1 and 2 before and after OCI at 1250 Hz **d** and 1350 Hz **e**, respectively. All data are obtained using a short-time Fourier transform and only the two frequencies of interest are shown.

by $2.11 \pm 0.39$ folds, whereas the energy involved in cross-talk is reduced to $0.0070 \pm 0.0025$.

## Frequency-multiplexed channel conditioning

The success of our approach opens a myriad of possibilities for controlling acoustic communications. Our ARMs can modulate the reflective phases over a broad frequency range. Such a capability enables broadband or frequency-multiplexed control. For example, by using a different objective function $\mathcal{G}_2(\mathbf{H}) = \left[ N - R_{\text{eff}}(\mathbf{H}) \right] + w_2$, where

$$w_2 = \frac{\sum_{i+j \neq N+1} |h_{ij}|}{\sum_{i+j=N+1} |h_{ij}|},$$

we can obtain a different kind of OCI: $\mathbf{H}$ is maximized in channel capacity, but it takes an anti-diagonal form. For 2-channel cases, it means that microphone 1(2) now only detects signal from loudspeaker 2(1). In addition, we further leverage the bandwidth of the ARMs to achieve frequency-multiplexing of the channels. For example, in Fig. 4, we show that the $2 \times 2$ channel matrices are simultaneously minimized for $\mathcal{G}_1$ at 1250 Hz, and for $\mathcal{G}_2$ at 1350 Hz. Because the frequency separation is far greater than the coherence bandwidth of the room, this essentially requires the simultaneous control of two

independent sets of degrees of freedom (cavity modes), which is far more challenging than the single-frequency scenario shown in Fig. 3. Figure 4a, b plot the channel matrices and the objective functions, wherein the two matrices clearly take diagonal and anti-diagonal forms after the optimization, respectively. The auditory effect of the optimization is further confirmed in Fig. 4c–e. The two loudspeakers emit temporally separated beeps in succession with two peaks in the Fourier domain, 1250 and 1350 Hz (Fig. 4c). When the different OCI are simultaneously attained, microphone 1 detects the first (second) beep but only picks the 1250-Hz (1350-Hz) components, whereas microphone 2's detection is inversed. These results are in stark contrast to the cases without OCI, for which two microphones always detects two beeps (Fig. 4d, e). In terms of signal intensities at the two frequencies, it is evident that the desired signals are improved, and the unwanted signals are effectively suppressed (data marked in Fig. 4d, e).

Leveraging the multi-frequency OCI, we are able to achieve the simultaneous crosstalk-free playback of two different pieces of music from the two separate sources. The results are summarized in Fig. 5 and are presented in Supplementary Movie 1. In this experiment, we selected two music pieces: "The C-D-E Song" and "Hot Cross Buns"

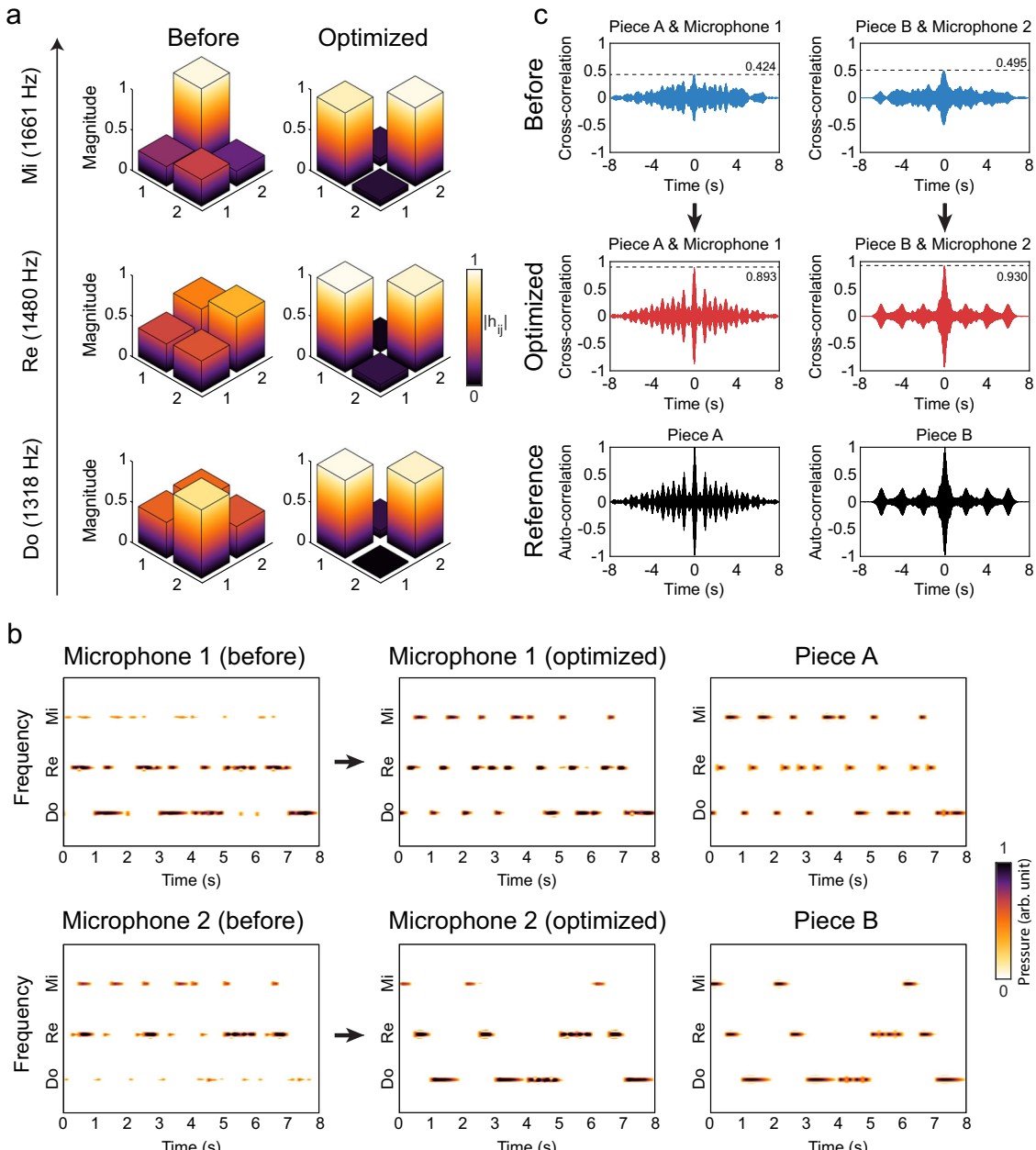

**Fig. 5 | Crosstalk-free simultaneous music playback from two sources. a** Three 2 × 2 channel matrices, each for a music note (frequency indicated), are simultaneously optimized for maximal effective rank and degree of diagonalization. **b** shows the detected audio signals from two separated microphones when two music pieces are simultaneously played from two loudspeakers in the room. In the unoptimized case (left column), the signals from the two sources are heavily mixed for both microphones. In the optimized case (middle column), both microphones receive the clean signals that are intended for them. As a reference, the original music signals are plotted in the right column. **c** The comparison of the cross-correlations between of the experimentally detected signals and the original music. The cross-correlations are significantly increased by the optimization. The bottom row is the auto-correlations of the two music pieces as a reference.

(Piece A and B in Fig. 5, respectively), both consisting of three identical musical notes: $f_{do}$ = 1318 Hz, $f_{re}$ = 1480 Hz, $f_{mi}$ = 1661 Hz, Three independent 2 × 2 channel matrices, each representing the channels for one note, are simultaneously optimized by the ARMs to minimize $\mathcal{G}_3$, given by

$$\mathcal{G}_3 = \left\{ 2 - \frac{1}{3} \sum_{x}^{do,re,mi} R_{eff}\left[\mathbf{H}(f_x)\right] \right\} + \frac{1}{3} \sum_{x}^{do,re,mi} w_1\left[\mathbf{H}(f_x)\right]. \quad (4)$$

In Fig. 5a. we can see that the optimization can indeed produce three near-full-rank channel matrices in diagonal forms. Prior to the optimization, the two music pieces played from the two loudspeakers

and received by the two microphones overlap in both frequency and time domains, as shown in Fig. 5b (left column). For human ears, the two pieces are heavily mixed and indistinguishable. After the optimization, the two microphones can each pick up only the piece that is intended for each of them. The received spectral-temporal signals are almost identical to the corresponding original music piece, as shown in Fig. 5b (middle and right columns). To further benchmark the results, Fig. 5c plots the cross-correlation functions between the audio signals received by the two microphones prior to and after the optimization. The peak values of the cross-correlation functions are raised from 0.424 and 0.495 to 0.893 and 0.930 for the two microphones, respectively. Moreover, the post-optimization cross-correlations are significantly improved and are nearly identical to the auto-correlations

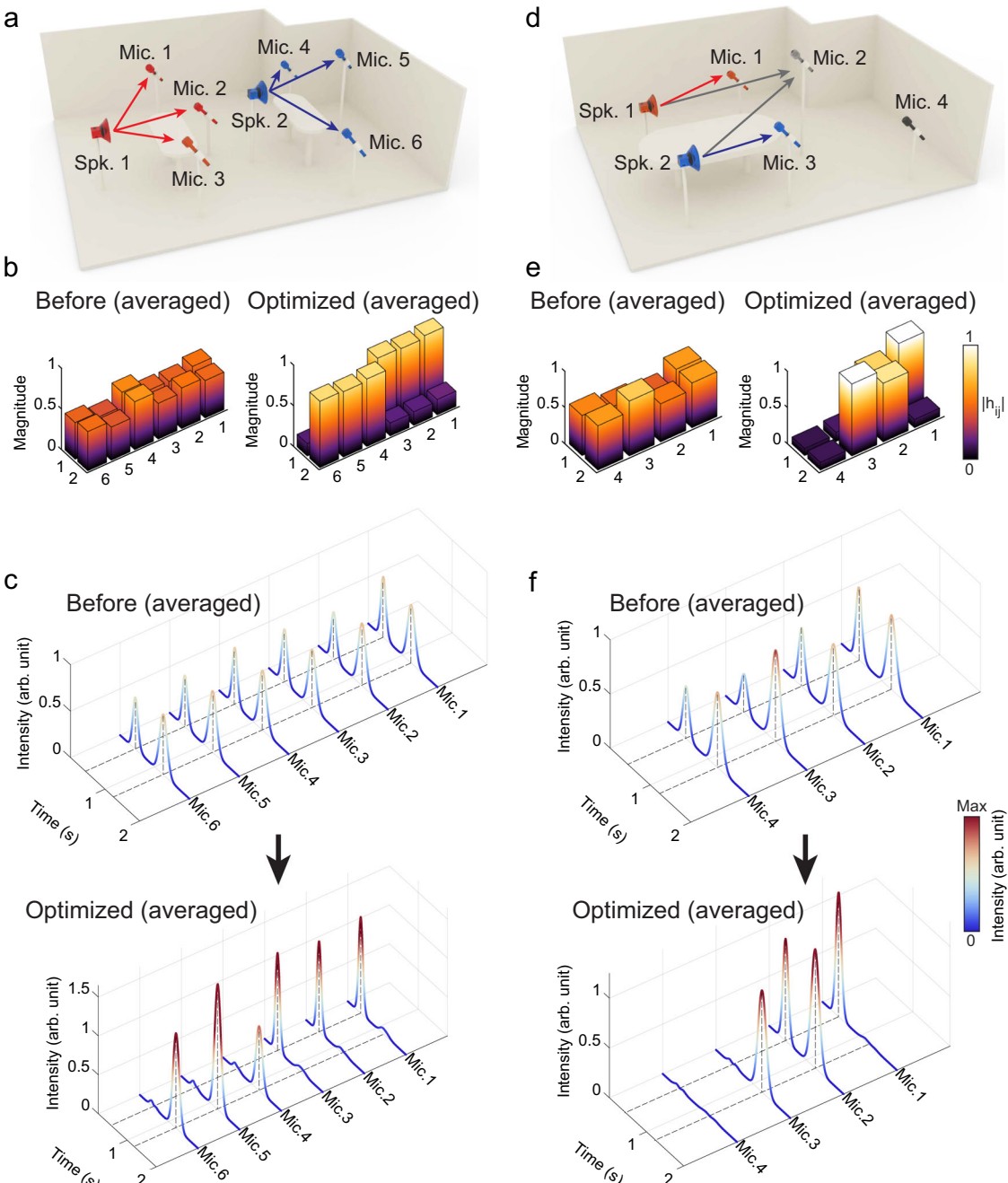

**Fig. 6 | Two different multi-user channel conditioning scenarios. a–c** A configuration of two loudspeakers (Spks.) and six microphones (Mics.), which is described by 6 × 2 channel matrices. **a** The desired effect is to separate the microphones into two groups, each group detects the sound from only one loudspeaker. **b** Channel matrices before (left) and after (right) the optimization. **c** The time-domain signals received by microphones 1–6. The desired signals are enhanced by an average of 2.9 dB, and unwanted signals are suppressed by 9.7 dB. **d–f** A configuration of two loudspeakers and four microphones, described by 4 × 2 channel matrices. **d** shows the desired effect. **e** Channel matrices before (left) and after (right) optimization. **f** The time-domain signals received by microphones 1–4. The desired signals are enhanced by 2.2 dB on average, and the unwanted signals are efficiently reduced by 15.7 dB.

of the two original music pieces, as shown in Fig. 5c. Please view Supplementary Movie 1 to listen to the recorded audio effects of this experiment.

The operating bandwidth of the ARMs also enables OCI over a continuous frequency band. An experimental example is shown in Supplementary Fig. 8.

We remark that the multi-frequency demonstrations are far more challenging to achieve compared to the single-frequency OCI. Because the frequency separations are far greater than the coherence bandwidth of the room, the wavefields are completely uncorrelated and thus the ARMs essentially need to simultaneously control multiple independent degrees of freedom.

## Flexible multi-user channel conditioning

We next demonstrate the versatile capability of our scheme by studying two cases with unequal numbers of loudspeakers and microphones, which are rather common scenarios. The first example, with two loudspeakers and six microphones, is shown in Fig. 6a–c. This is a type of configuration that often emerges, e.g., group discussions in a shared office. The channel matrix, in this

case, is $6 \times 2$ in dimensions, and the upper bound of $R_{eff}$ is 2. We impose the demand that the microphones are separated into two groups, and each group is tuned in to only one loudspeaker, i.e., microphones 1–3 (4–6) wish only to capture loudspeaker 1(2), and ignore loudspeaker 2(1). By using the proper objective function, a channel matrix that suits the need is successfully produced, as shown in Fig. 6b. Similar to the above experiments, we send two beeps separated in time with the two loudspeakers. The signals received by the six microphones at different positions are shown in Fig. 6c, wherein it is clearly seen that the intended auditory effect is successfully achieved. Specifically, the desired signals are enhanced by an average of 2.9 dB, and unwanted signals are efficiently reduced by 9.7 dB.

In the second example shown in Fig. 6d–f, the configuration is described by a $4 \times 2$ channel matrix. To make an interesting case, we impose a complicated set of "demands": microphone 1(3) only picks up loudspeaker 1(2), microphone 2 needs to detect both loudspeakers, and microphone 4 is shielded from all sources. By using the proper objective function, the desirable channel matrix is indeed obtained, as shown in Fig. 6e. Similar to the above experiments, we send two beeps separated in time with the two loudspeakers. The signals received by the four microphones at different positions are shown in Fig. 6f, wherein it is clearly seen that the intended auditory effect is successfully achieved. Specifically, the desired signals are enhanced by 2.2 dB on average, and the unwanted signals are efficiently reduced by 15.7 dB.

## Discussion

By a successful crossover of multiple scattering media, adaptive wavefield shaping, acoustic metasurfaces, and communication theories, we have achieved effective control of complex acoustic waves. The properties of channel matrices in disordered wavefields play a crucial role. Unlike scattering matrices for multiple scattering media, the channel matrices are typically small-sized random matrices. Their dimensionality is determined not by the complexity of the medium but by the number of sources and receivers. Hence, they obey different sets of statistical distribution laws compared to large-sized random matrices, in which the distribution of singular values can be derived from the Marčhenko-Pastur law[27] (for square matrices, it becomes the quarter-circle law[28]). By using random matrix theory and probability theory, the statistical distribution of key parameters, such as the effective rank, can be obtained numerically and theoretically, and they agree well with the experimental results. The relevant analyses and results are presented in Supplementary Note 5.

The wavefield modulation is achieved by the ARMs. Compared to the previous modulating device based on membrane-type acoustic metasurfaces[14], the ARMs used here are more advanced in several important ways. First, they modulate the phase of the reflected waves instead of the transmitted waves, which means that it functions by altering the boundary conditions of the room. This implies that the implementation of ARMs requires less modification to the interior space, which is desirable for most real-life applications. Second, the functional bandwidth is significantly improved, which is not only advantageous for broadband or frequency-multiplexed applications but also beneficial to coherent control of time-varying sound. It is possible to enlarge the bandwidth by further tailoring higher-order resonant modes of building blocks or by combining panels with different working frequencies. Third, they contain no soft elastomer parts, which makes them far more reliable and durable. We also remark that the ARMs should not be considered as diffusers. Its function is not to scatter waves evenly in all directions for the formation of a uniform wave field. Instead, it scatters waves in specific ways designed to intentionally disrupt an already uniform reflected wave field, thereby achieving OCI.

Our approach achieves channel isolation through the physical modulation of the complex sound field. This is unlike any traditional strategy that often relies on restricting the sources or the receivers[29–31], e.g., putting on a noise-blocking headsets. This research highlights that modifying the channel matrix during the cross-talk cancellation process can be an effective approach, offering new solutions and technological means in related fields.

In practical scenarios of acoustic communications, the positions of sources and receivers are often interchanging. When the numbers of sources are equal to the receivers, i.e., the channel matrix is a square matrix, the system has reciprocity once OCI is achieved. In other words, no further optimization is required to handle the exchange of sources and receivers. However, if the numbers of sources and receivers are different, the corresponding channel matrix does not respect reciprocity. For more detailed discussion, please refer to Supplementary Note 1.

Our method relies on moderate reverberation. Therefore, for optimal performance, the sources and the receivers shall be greater than the reverberation radius (0.5 m in the existing experimental configuration). If reverberation is weak or even completely absent, direct sound dominates and the effectiveness of our method is compromised. (For instance, this experiment cannot be conducted in an anechoic chamber.) On the contrary, excessively long reverberation time also affects performance by increasing the correlation between the optimal states of two different ARMs, resulting in a reduction in the number of controllable modes and consequently compromising the performance of the reflector.

There are several routes that can potentially improve the overall performance of our channeling conditioning approach. First, our results are achieved using a rudimentary climbing algorithm and without prior knowledge of the acoustic environment (other than some of its basic properties). We anticipate that more advanced optimization algorithms can lead to better results and reduce the optimization time. Imaging techniques such as phase conjugation, inversed filtering, together with prior knowledge of the acoustic environment, are also viable routes for improving the outcomes[32]. Second, better results are expected if the phase modulation is of a finer phase sampling rate, e.g., a four-phase modualtion[33]. However, this is at the cost of longer optimization time. This is readily achievable using our current ARMs, but at the cost of long converging time. Finally, other active acoustic designs are potentially suitable for achieving similar functionalities in sound-field manipulations[34–37]. In summary, we have demonstrated the flexible control of the acoustic wave properties in cavities for versatile acoustic communication needs. Our results have immense potential towards next-generation smart acoustic technologies that may revolutionize how we manipulate, perceive, and experience sound. It may also inspire new technologies in vibration controls, ultrasonics, etc., and open new possibilities for manipulating wave scattering and wave chaos.

## Methods

### The properties of the experimental environment

The experiment was conducted in an irregularly shaped room with furniture inside. The volume of the room is $V \approx 44\,\mathrm{m}^3$, and the total surface area is $A \approx 78\,\mathrm{m}^2$ (Fig. 2a). From the averaged acoustic impulse responses[38], the reverberation time for a 60-dB decay is found to be $T_{60} \approx 0.52\mathrm{s}$[38,39]. The spatial standard deviations of the sound pressure level in the room are experimentally characterized to be 0.655 dB in 1000–2000 Hz, and 0.562 dB in 250–8000 Hz[40]. A more detailed discussion can be found in Supplementary Note 2. Using this value, the Schroeder frequency is $f_S = 2000\sqrt{T_{60}/V} \approx 217\mathrm{Hz}$, which is much lower than the experimental frequencies. The exponential decay time of the room is $\tau = T_{60}/\ln 10^6 \approx 38\mathrm{ms}$, which leads to a coherence bandwidth of $f_{co} = (\pi\tau)^{-1} \approx 8.4\mathrm{Hz}$. The modal density at frequency $f$ is given by $N(f) \approx (\frac{4\pi V}{c^3}f^2 + \frac{\pi A}{2c^2}f)f_{co}$, with $c = 343\mathrm{m/s}$ being the speed of sound, so the modal density ranges from ~149 at 1100 Hz to ~410 at 1850 Hz[38].

Using numerical simulations (the ray acoustics module of COMSOL Multiphysics), we estimate the scattering mean free path $\ell \approx 1.27$ m, so the mean interval between two scattering events for a wave is $\Delta t_s = \ell/c \approx 3.7$ ms. Hence, a sound wave, on average, undergoes roughly 10 scattering events before it decays, such that the resulting field is speckle-like. The spatial distribution of the field amplitude follows Rayleigh distribution, which means the room is a chaotic cavity[41,42]. By applying the central limit theorem, the real and imaginary parts of the pressure both follows the Gaussian distribution, so the pressure amplitudes conform to the Rayleigh distribution[43]. We have confirmed such properties of the sound field by experimentally raster-scanning multiple planes of the sound fields in the room, as shown in Supplementary Fig. 4. We remark that the distribution is valid for most locations in the room, except for the immediate neighborhood of the source (within the reverberation radius), in which the direct sound dominates, and the assumption of ray i.i.d. is not satisfied. Therefore, all experiments are conducted with microphones outside the reverberation radius.

### The properties of the channel matrix

The entries of the channel matrix follow the same statistical distribution as the sound field, hence they are modeled using complex random numbers. The singular values of such matrices are not uniform, and in particular, for large random matrices, the distribution of singular values can be derived from the Marčhenko-Pastur law[27]. Also, the channel matrix need not possess any symmetry (such as transposition or Hermitian conjugation). Numerically, we generate the real and imaginary parts of each entry of the channel matrix as Gaussian random numbers. A total of 10,000 different channel matrices are numerically produced and their properties, including $R_{\mathrm{eff}}$, $w_1$, and (or) $w_2$ are recorded for comparison with the experimental values. For $2 \times 2$ channel matrices, the average $R_{\mathrm{eff}}$ is about 1.7, and $w_1$ is about 1.2, which are indicated in Figs. 3c and 4b.

### The design and characterization of the ARMs

The ARMs are based on reflective acoustic metasurfaces that are set against the walls of the room. They are essentially tunable boundaries of the room as an acoustic cavity. The ARMs consist of a square array of identical THRs. There are a total of 200 independent units of THRs. Figure 2a illustrates the design of the THR, including its dimensions. The natural frequency of the Helmholtz resonance is tunable by changing the volume of the belly. Specifically, a small stepper motor is used to turn a hoop, which can be rotated between two positions. The stepper motors are controlled by Arduino Mega 2560 boards programmed by MATLAB. At the open position, the belly is a cuboid. At the closed position, the partition on the hoop and the internal partitions in the belly form a cylinder with a smaller volume. The reflection of a single THR is characterized using an acoustic impedance tube. The natural frequency of the Helmholtz resonance is found to be $f_o = 990$ Hz for the open state, and $f_c = 1650$ Hz for the closed state, such that the two states generate a difference of 140–160° in the reflection phase difference (Fig. 2b).

In the demonstration shown in the Supplementary Movie 1, the upper bound of the working frequency range of the ARMs is expanded to roughly 2000 Hz. This is achieved by taking the second-order resonance of the THR into consideration, which is at 2010 (3410) Hz in the open (closed) state.

### Experimental procedures

The sources (loudspeakers) and receivers (microphones) are placed in different positions in the room under three constraints. First, their mutual separation is larger than the correlation length, which is about half the wavelength. Second, the microphones and the loudspeakers are separated by at least 1.5 m, which is larger than the reverberation radius ($\sim 0.5$ m). Third, for the same set of experiments, the distance between the microphones and loudspeakers are roughly the same for different configurations such that the pulses arrive at roughly the same time. This is to ensure that the temporal signals in each realization roughly overlap so that the averaging process is well-defined. (Note that this condition is imposed not for channel conditioning, but for the ease of data processing.) For different configurations, the positions of the loudspeakers and the microphones are changed by at least half a wavelength. Respecting these three constraints, the changes are as random as possible. The loudspeakers and microphones are connected to NI-cDAQ-9174, with NI 9260 as outputs and NI 9234 as inputs. The device is controlled by a PC.

The modification to the channel matrix is achieved by feedback-driven optimizations based on a climbing algorithm. The channel matrix is measured at each step and sent to the controlling PC. The PC computes the relevant parameters, such as the effective rank, then the objective function. Then, the program instructs up to 15 randomly chosen THRs to switch the states, then the channel matrix is measured again. The process is repeated until the objective function converges to the target value. Please refer to Supplementary Note 9 for a detailed algorithm procedure description.

At the current stage, the optimization of a $2 \times 2$ channel matrix at a single frequency typically takes 2–5 minutes. For more complex scenarios, the optimization time will inevitably be longer. The main limitation is the time required for switching the states of control circuits and mechanical structures. To overcome this issue, potential improvements include using advanced control circuits like FPGA for better performance and refining the mechanical parts for faster state switching. More intelligent optimization algorithms can also be applied to reduce optimization time.

## Data availability

The data that generate the results of this study are available from the corresponding authors upon request.

## Code availability

The codes supporting the findings of this study are available from the corresponding authors upon request.

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

## Acknowledgements

This work is supported by the National Key R&D Program of China (2022YFA1404400), the National Natural Science Foundation of China (11922416), and the Hong Kong Research Grants Council (RFS2223-2S01, 22302718, A-HKUST601/18). M.F. acknowledges partial support from the Simons Foundation/Collaboration on Symmetry-Driven Extreme Wave Phenomena.

## Author contributions

G.M. initialized and supervised the research. H.Z. and Q.W. performed the experiments. Q.W. designed and fabricated the ARMs and H.Z. performed the wavefield shaping. H.Z. analyzed the random sound fields. All authors analyzed the data. H.Z. and G.M. wrote the manuscript with inputs from Q.W. and M.F.

## Competing interests

The authors declare no competing interests.
