## [Peer Review File · Nature Communications]

Optimizing multi-user indoor sound communications with acoustic reconfigurable metasurfacesReviewer #1 (Remarks to the Author):

Report on NCOMMS-23-34970

This manuscript reports the use of 200 electronically-controlled adjustable Helmholtz resonators to form parts of the room walls, with the aim of manipulating the sound field in such a way that in a multiple sound source environment, any source/receiver pair can clearly communicate with each other without crosstalk. The authors have demonstrated the effectiveness of this approach by using an algorithm based on optimizing the Shannon entropy of the channel matrices.

The work clearly demonstrates the potential of active acoustic metamaterials in achieving previously unthinkable effect; even though I question the practicality of its actual application, for the reasons I present below. However, as a work that illustrates the potential of active acoustic metamaterials, I still recommend its publication in Nature Communications, as it might stimulate further developments towards actual applications.

I have a few questions and comments that the authors should consider, preferably with reasonable responses, before the manuscript can be published.

1. Is there reciprocity in the source/receiver relation? In other words, for practical communications, it is most often that the source/receiver relation is reversed in a two-way conversation. Hence reciprocity is the most important, in contrast to the examples presented where the receiver is only listening to a single frequency sound or some music. In my view, as the source sound has to bounce multiple times before reaching the receiver, changing the location of the source may required a renewed optimizing process. Is that true?

2. How long does it take to optimize the target function each time? Even though the authors have mentioned that it would take longer for the multi-frequency case to achieve the optimality, the actual elapsed time is not mentioned in the text. The authors should make such information explicit, as this might be a crucial bottleneck for future applications.

3. In the first example of a single frequency, 2 by 2 channel matrix, it seems that the numerical optimization can achieve a diagonal channel matrix. Doesn't that mean the same can be achieved numerically through diagonalization of the matrix? If so, the authors should state it.

Reviewer #2 (Remarks to the Author):

The manuscript presents a practical framework for ensuring accurate communications between multiple sources and receivers in a reverberant environment, avoiding crosstalks by combining communication theory, optimization, and wave scattering of discrete arrays of tunable Helmholtz resonators. It is then a rather complex problem, spanning over different scientific fields that seems a bit challenging to synthesis, although I acknowledge the authors have been able to make it legible. To my opinion, the most important and meaningful result relies on the formulation of ad hoc cost functions for improving the communication channels in the room, the rest being somewhat at the level of the state of the art.

The manuscript is generally well written, except a few poor formulations here and there. The structure is easily legible and the figures fairly support the interpretations.

However, although the application of tunable Helmholtz resonators, and their optimization for multi-channel communication improvement in reverberant fields seems somehow new (apart from the prior references of the authors, I don't have nailed any other such examples), my global impression is that the work reported here does not present enough novelty neither substantial practical results to the fields covered by Nature Communications for the following reasons:

- the proposed "tunable" (in fact, only switchable between two states) Helmholtz resonator concept is not at the forefront of the state of the art of active diffusers, and is too specific to a concept formerly proposed by the same authors/institutions. Alternatives active solutions known to provide far more tunability are not even mentioned in the manuscript. With the proposed solution, it is only possible to achieve a binary change of acoustic properties, resulting in two specific operating frequencies, whereas other solutions could actually present more broadband acoustic properties. I understand the motivation of the authors (their in-house concept is an asset of their

group that is readily available for experimental assessments) but this does not really represent the best tool for the proposed application. Note that the proposed embodiment differs technically from the ones already reported in the literature, but this does not justify any novelty in terms of concept.

- moreover, still on the physical aspects of the paper, the "metasurface" label seems off-topic in this work, besides pure "marketing".
- the argument on "reverberant environment": I doubt the practical example shown in this paper can extend beyond this relatively poorly reverberant environment: with the reverberation times reported in the Methods section ($T_{60} \sim 0.52$ s), I doubt that the tested room can be qualified as "reverberant". It is especially obvious in Figures 3d, 4c,d,e and 6c,f, where the recorded signals show the dryness of the room... Moreover, it would have been interesting to show how the performance degrade with reverberation time in the room.
- on the argument on broadbandness: 40 Hz at 1'300 Hz represents only 1/24th octave, which is far from a broad frequency range. The authors should be very prudent with such claims ...
- I am not fully convinced by the demonstration, focusing on pure tones at discrete frequencies. This doesn't make a sound demonstration of application with respect to (speech) communication problems: what happens with real speech signals instead, or at least with band-limited modulated noises? This is not even mentioned in the paper, nor in the conclusions.

To summarize, the work reported in this manuscript shows some interesting ideas and results, that performs accordingly to the theory but only for too limited acoustic signals, and in a rather "non-reverberant" situation. Then it fails to demonstrate an actual solution that could be representative of a real-life situation. Consequently and although the potential applicative outcomes could have been very interesting, it does not contain ground-breaking results and therefore it is not suited, in my opinion, for publication in a journal such as Nature Communications.

Reviewer #3 (Remarks to the Author):

This manuscript introduces a new approach to enhance multi-user sound communications in reverberant environments. The primary focus of this method is the mitigation of crosstalk between multiple talkers, facilitated by adaptable binary acoustic metasurfaces strategically positioned on room walls. The underlying challenge addressed in this study involves the interference arising from reverberation when multiple individuals speak simultaneously. Despite humans have the ability to selectively attend to specific speakers, known as the cocktail party effect, enhancing speech perception by minimizing crosstalk remains advantageous, which shows the scientific significance of this work.

Conventional strategies typically target crosstalk reduction at either the source (loudspeakers) or the receiver (microphones). For instance, cancellation filters are employed to attenuate crosstalk between loudspeakers in binaural or multi-channel audio setups, as discussed in A. Roginska, and P. Geluso (Eds.), *Immersive Sound: The Art and Science of Binaural and Multi-Channel Audio*, (London, Routledge, Taylor & Francis Group, 2018), and B. Xie, *Spatial Sound: Principles and Applications*, (Boca Raton, CRC Press, 2023). Signals captured by microphones are processed to extract distinct audio signals from each source, see in J. Benesty, M. M. Sondhi, and Y. Huang (Eds.), *Springer Handbook of Speech Processing*, (London, Springer, 2008). However, this study presents a new proposition: the elimination of crosstalk among transfer functions connecting sources and receivers. While the work has the potential to serve as a valuable source of inspiration for researchers, certain limitations need to be addressed, as outlined below:

1. I suggest authors mention the cocktail party effect in the Introduction. For example, see the reference B. Arons, "A review of the cocktail party effect," *Journal of the American Voice I/O society* 12(7), 35–50 (1992).

2. The current performance evaluation focuses on triple tones (Do Re Mi), which may not accurately reflect real-world scenarios where speech signals are often broadband (typically at least ranging from 500 Hz to 4 kHz). The effectiveness of the proposed system for wideband signals remains uncertain and merits further investigation/discussion.

3. The optimization of crosstalk cancellation is centered around specific frequencies, leading to impressive results in some instances (e.g., Fig. 3a). However, it is likely that this targeted optimization could negatively impact performance in other frequency ranges, possibly adversely enhance crosstalk in those frequency ranges. To provide a comprehensive understanding, it is recommended that the authors present channel isolation metrics across a wider frequency spectrum (500 Hz to 4 kHz at least).

4. In audio engineering, presenting results in decibels (dB) instead of a linear scale (e.g., Fig. 4(c-e)) would offer a more meaningful representation, aiding readers in grasping the magnitude of improvements with greater clarity.

Summary of major revisions:

Revisions	Location	Addressing
Added the discussion on the reciprocity of the channel matrix.	Supplementary Note 1 and Supplementary Fig. 1; Main text, Discussion (Page 8).	Reviewer 1
Added optimization time and discussion on potential improvement directions.	Main text, Methods (Page 11).	Reviewer 1
The description of the channel isolation index in the manuscript has been enhanced.	Main text, Results (Page 4).	Reviewer 1
Differentiate the acoustic metasurfaces with diffusers.	Main text, Discussion (Page 8).	Reviewer 2
Additional experimental results and discussion on the reverberant nature of the room and relevant references are included.	Supplementary Note 2 and Supplementary Fig. 2; Main text, Methods (Page 9); Ref. [35,36].	Reviewer 2
Added the discussion on the degree of reverberation on the performance.	Main text, discussion (Page 8).	Reviewer 2
The discussion and figures in the manuscript have been revised to prevent confusion or overstatement. For example, dummy persons in Figs. 3 and 6 have been replaced with speakers and microphones. The wordings “speakers” and “listeners” are replaced with “loudspeakers” and “microphones.”	Main text, Fig. 3, 4, 6.	Reviewers 2, 3
The use of “broadband” is removed. The continuous-frequency experiment now shows the successful results of 100-Hz bandwidth, which is larger than the 40-Hz bandwidth result in the first submission.	Supplementary Note 8 and Supplementary Fig. 8.	Reviewer 2, 3
A discussion on the cocktail party effect has been included and relevant references are included.	Main text, introduction; Ref. [23].	Reviewer 3
A discussion on traditional strategy to improve acoustic communication has been included. Relevant references are included.	Main text, discussion (Page 8), Ref. [29,30,31].	Reviewer 3
Added experimental data on the channel isolation index within the 500-4000 Hz range.	Supplementary Note 6 and Supplementary Fig. 7; Main text (Page 5).	Reviewer 3
Added corresponding decibel values in Figures.	Main text, Fig. 3, 4.	Reviewer 3

Reviewer #1 (Remarks to the Author):

This manuscript reports the use of 200 electronically-controlled adjustable Helmholtz resonators to form parts of the room walls, with the aim of manipulating the sound field in such a way that in a multiple sound source environment, any source/receiver pair can clearly communicate with each other without crosstalk. The authors have demonstrated the effectiveness of this approach by using an algorithm based on optimizing the Shannon entropy of the channel matrices.

The work clearly demonstrates the potential of active acoustic metamaterials in achieving previously unthinkable effect; even though I question the practicality of its actual application, for the reasons I present below. However, as a work that illustrates the potential of active acoustic metamaterials, I still recommend its publication in Nature Communications, as it might stimulate further developments towards actual applications.

I have a few questions and comments that the authors should consider, preferably with reasonable responses, before the manuscript can be published.

Response:

We thank the reviewer for his/her careful review and positive remarks on our work and the recommendation for publication. Below we present point-to-point answers to the comments.

1. Is there reciprocity in the source/receiver relation? In other words, for practical communications, it is most often that the source/receiver relation is reversed in a two-way conversation. Hence reciprocity is the most important, in contrast to the examples presented where the receiver is only listening to a single frequency sound or some music. In my view, as the source sound has to bounce multiple times before reaching the receiver, changing the location of the source may require a renewed optimizing process. Is that true?

Response:

Thank you for this very interesting question. The simple answer is: yes, when the numbers of sources and receivers are the same, i.e., the channel matrix is a square matrix, reciprocity exists since the channels are nearly independent after optimization. In this case, exchanging the positions between sources and receivers does not require to perform the optimization again. But when the numbers of sources and receivers are different, reciprocity among them may break.

To show this, we first establish that the room as an acoustic cavity is reciprocal. Sound propagation inside the cavity is governed by an acoustic equation that is a quadratic differential equation in both space and time. Therefore, both spatial reciprocity and time-reversal symmetry hold. The inevitable presence of dissipation does slightly break time-reversal symmetry but not to the degree that it breaks reciprocity. In fact, we can perform high-quality acoustic time-reversal experiment to re-focus sound back to source position in this room. Similar experiments have been reported before, e.g., in Ref. 1 and 2 (in different rooms but there is no fundamental difference in terms of the physics).

Square matrix, channels coupled

Non-square matrix

Square matrix, isolated channels

Fig. R1 On the reciprocity of channel matrices. **a.** Generically, the reciprocity condition requires the channel matrix to have transpose symmetry. **b.** For square channel matrices with isolated channels, reciprocity holds. **c.** Reciprocity is not satisfied when the channel matrix is not a square matrix.

Coming back to the channel matrix, the reciprocity condition requires the channel matrix to have **transpose symmetry**, i.e., $\mathbf{H} = \mathbf{H}^T$ (Fig. R1a). In general, this condition is not satisfied, because there is no guarantee that $h_{12} = h'_{12}$ and $h_{21} = h'_{21}$. However, after channel isolation is attained, the channel matrix becomes a near-diagonal matrix with $h_{12} \cong h'_{12} \cong 0$, and $h_{21} \cong h'_{21} \cong 0$ so that the matrix is near symmetric, so reciprocity is largely satisfied (Fig. R1b, left). The same clearly also holds for near anti-diagonal channel matrix (Fig. R1b, right). Hence, there is **no need** for re-optimization when exchanging the positions of the sound sources and the receivers. These correspond to the cases in Fig. 3 and Fig. 5 in the main text.

However, in scenarios where the numbers of sources and receivers are unequal, re-optimization of the room configuration is required, as shown in Fig. R1c and Fig. 6 in the main text.

Taking this valuable advice into account, appropriate modifications have been made to the text to reflect this aspect accordingly. Specifically, we have included a discussion of this aspect in the revised Supplementary Information, labeled as Supplementary Note 1. Furthermore, it has been added to the Discussion section:

“In practical scenarios of acoustic communications, the positions of sources and receivers are often

interchanging. When the numbers of sources are equal to the receivers, i.e., the channel matrix is a square matrix, the system has reciprocity once OCI is achieved. In other words, no further optimization is required to handle the exchange of sources and receivers. However, if the numbers of sources and receivers are different, the corresponding channel matrix does not respect reciprocity. For more detailed discussion, please refer to Supplementary Note 1.”

2. How long does it take to optimize the target function each time? Even though the authors have mentioned that it would take longer for the multi-frequency case to achieve the optimality, the actual elapsed time is not mentioned in the text. The authors should make such information explicit, as this might be a crucial bottleneck for future applications.

Response:

We thank the reviewer for the suggestion. In our present experimental setup, the average duration for optimizing a 2x2 channel matrix at a single frequency is 2-5 minutes. For other more complex scenarios, such as those in Fig. 5, the minimum condition is far more stringent, so it may take up to 1 hour.

As a proof of concept, we did not target the time performance of the optimization algorithm. The optimization algorithm is a simple random iteration method. We believe that the optimization time can be further shortened by several means. For example, more advanced control circuits, such as FPGA, which offer much better performance compared to Arduino (which cannot send and receive signals at the same time). Also, there is plenty room to optimize the mechanical mechanism for faster state switching – right now, the mechanism is built for reliability and each state switching takes around 2 seconds. Additionally, more intelligent optimization algorithms, such as the ant colony algorithm, holds promise for effectively reducing the optimization time.

We added a note on optimization time to the Discussion section of the main text:

“At the current stage, the optimization of a 2×2 channel matrix at a single frequency typically takes 2-5 minutes. For more complex scenarios, the optimization time will inevitably be longer. The main limitation is the time required for switching the states of control circuits and mechanical structures. To overcome this, potential improvements include using advanced control circuits like FPGA for better performance and improving the mechanical structures for faster state switching. More intelligent optimization algorithms can also be applied to effectively reduce optimization time.”

3. In the first example of a single frequency, 2 by 2 channel matrix, it seems that the numerical optimization can achieve a diagonal channel matrix. Doesn't that mean the same can be achieved numerically through diagonalization of the matrix? If so, the authors should state it.

Response:

We appreciate this question. First of all, we remark that all the data shown in the main text figures are experimental results instead of numerical simulations. There is simple answer to the reviewer's question: the channel matrix is not always guaranteed to be diagonalizable. Hence, we opt for an alternative route that considers information entropy that relates to singular-value decomposition of the channel matrix, which is always doable. This method is also advantageous in its wide applicability, e.g., it enables us to deal with situations in which the channel matrix is rectangular (results in Fig. 5). We have conducted additional analysis in the Supplementary Information Note 3 of our initial submission (Note 4 in the revised one), and we present the main results here.

In addition, even when consider 2-by-2 channel matrix, direct diagonalization is not the optimal method. For example, it is entirely possible that the two eigenvalues, denoted λ_1 and λ_2 , are drastically different

(e.g., $\lambda_1 = 1$, $\lambda_2 = 0.01$). This means that, although the two eigenchannels are found, the second eigenchannel (λ_2) offers very low signal-to-noise ratio compared to the first eigenchannel, which is clearly not optimal for communications. However, this situation is automatically avoided when considering effect rank $R_{\text{eff}}(\mathbf{H})$, defined as

$$R_{\text{eff}}(\mathbf{H}) = \exp\left(-\sum_{k=1}^N p_k \ln p_k\right) \leq \exp\left[N \ln\left(\frac{1}{N} \sum_{k=1}^N p_k^{-p_k}\right)\right] \leq N, \quad (\text{R1})$$

where $p_k = \sigma_k / (\sum_{i=1}^N \sigma_i)$ are the normalized singular values of \mathbf{H} . It can be shown that the equal sign holds when all p_k are equal, i.e., $p_k = 1/N$. In other words, $R_{\text{eff}}(\mathbf{H})$ is maximized when the two singular values are nearly identical, which not only guarantees independent channels but also imposes the same signal-to-noise ratio for all channels.

In response to the reviewer's comments, we have added a note to the main text:

"We remark that the minimization of either R_{eff} or w_1 alone is insufficient for achieving OCI, and it is necessary to minimize both of them simultaneously. For example, minimizing w_1 alone, i.e., without enforcing a maximum R_{eff} , can still reduce off-diagonal entries. But there is no guarantee that the diagonal entries have near equal values. If the diagonal entries differ significantly, the two channels have drastically different signal-to-noise ratios, which is not optimal for communication purposes. For a detailed discussion on this issue and additional experiments, please refer to Supplementary Note 4."

We wish to also draw the reviewer's attention to the Supplementary Fig. 5 again (reproduced as Fig. R2 for convenience), wherein the role of the effective rank and degree of diagonalization are compared. It is seen that both factors need to be optimized for the best results.

Fig. R2 The roles of R_{eff} and w_1 in objective function $\mathcal{G}_1(\mathbf{H})$. **a** The magnitude-averaged entries of the channel matrix before the optimization. **(b, c, d)** correspond to the results obtained using $2 - R_{\text{eff}}(\mathbf{H})$, w_1 , and $\mathcal{G}_1(\mathbf{H}) = 2 - R_{\text{eff}}(\mathbf{H}) + w_1$, respectively.

Reviewer #2 (Remarks to the Author):

The manuscript presents a practical framework for ensuring accurate communications between multiple sources and receivers in a reverberant environment, avoiding crosstalks by combining communication theory, optimization, and wave scattering of discrete arrays of tunable Helmholtz resonators. It is then a rather complex problem, spanning over different scientific fields that seems a bit challenging to synthesis, although I acknowledge the authors have been able to make it legible. To my opinion, the most important and meaningful result relies on the formulation of ad hoc cost functions for improving the communication channels in the room, the rest being somewhat at the level of the state of the art.

The manuscript is generally well written, except a few poor formulations here and there. The structure is easily legible and the figures fairly support the interpretations.

Response:

We thank the reviewer for the positive comments. We have carefully studied the suggestions, and by addressing all of them, we believe the paper is significantly improved.

To begin, we would like to argue that the cost functions are clearly not *ad hoc*. The formulation of the cost functions rests on the systematic analysis of channel matrix, whose foundation is the theories of information and communications. Our formulation is general in that it can be applied to any reverberating environment and it can be used to handle any number of sources and receivers placed anywhere in the environment. Indeed, what we demonstrate here is not the general control of global acoustic properties but the refined control of the local sound fields for specific purposes, so the changes one has made is to account for the diverse communication scenarios. It clear does not mean the approach is *ad hoc*.

However, although the application of tunable Helmholtz resonators, and their optimization for multi-channel communication improvement in reverberant fields seems somehow new (apart from the prior references of the authors, I don't have nailed any other such examples), my global impression is that the work reported here does not present enough novelty neither substantial practical results to the fields covered by Nature Communications for the following reasons:

Response:

We thank the reviewer's comments and suggestions, which helped improve our manuscript. Reviewer raised concerns about the novelty of our study, and we provide some clarification here.

First, we are glad that the reviewer acknowledges that the acoustics effects reported in the manuscript have not been reported elsewhere, although we do agree that they still need improvement for handling real-life scenarios.

About the novelty of our work. We use information and communication theories to guide the active control of reverberating sound fields. This conceptually new approach has not been adapted for acoustics previously, to the best of our knowledge. In addition, we present the first demonstration of acoustic wavefield shaping for multi-frequency signals using passive reflective units, which has not been demonstrated before either.

- the proposed "tunable" (in fact, only switchable between two states) Helmholtz resonator concept is not at the forefront of the state of the art of active diffusers, and is too specific to a concept formerly proposed by the same authors/institutions. Alternatives active solutions known to provide far more tunability are not even mentioned in the manuscript. With the proposed solution, it is only possible to achieve a binary change of acoustic properties, resulting in two specific operating frequencies, whereas other solutions

could actually present more broadband acoustic properties. I understand the motivation of the authors (their in-house concept is an asset of their group that is readily available for experimental assessments) but this does not really represent the best tool for the proposed application. Note that the proposed embodiment differs technically from the ones already reported in the literature, but this does not justify any novelty in terms of concept.

Response:

First, the Helmholtz resonators we proposed should not be considered as active diffusers. Diffusers should ideally scatter waves evenly to all directions, such that a homogenous wavefield can be obtained. However, the functionality of our tunable metasurface is completely different: its goal is to scatter waves in specific ways such that the reverberating wavefield, which is already homogeneous, is purposely disrupted so the outcome is more suitable for multi-channel communications. In fact, in some particular realizations, it is possible for a good number of units to end up in the same state (Fig. R3), so the corresponding portion of metasurface produces near specular reflection, which is clearly different from any diffuser. In response to this comment, we added a note to the Discussion section:

“We also remark that the ARMs should not be considered as diffusers. Its function is not to scatter waves evenly in all directions for the formation of a uniform wave field. Instead, it scatters waves in specific ways designed to intentionally disrupt an already uniform reflected wave field, thereby achieving OCI.”

Fig. R3 Some examples where most units are in the same state. Black represents open state and white represents closed state.

Second, we would like to clarify that our metasurface consisting of Helmholtz resonators was designed and constructed in house specifically for this experiment, instead of utilizing any readily available assets in our lab. In fact, this is the first time that our group works on Helmholtz resonators. And the Helmholtz resonator is distinct in its tunable design from those reported in existing literature. In our previous works about wavefield shaping in reverberating rooms, the modulation was implemented by membrane-type acoustic metamaterials, which only offers modulation the transmitted sounds in <1 kHz regime^{3,4}. Hence,

they must be placed somewhere in the middle of the room. The device used in this work, however, modulates the reflected waves, so they can be regarded as tunable boundaries of the room.

Third, the music demonstration and the 40-Hz continuous band experiment (replaced by 100-Hz bandwidth experiment in revised version) already show that the capability of our two-phase-controlling reflector is not limited to two specific operating frequencies. The reflection phase can be actively tuned across a wide frequency range of 1100-1850 Hz, which is over 2/3 of an octave. This is shown by the black dashed curve in the upper panel of Fig. 2d in the main text.

Lastly, we are not aware of the “other solutions” mentioned by the reviewer. To the best of our knowledge, acoustic wavefield shaping in reverberating cavity has only been achieved using either the membrane-type metamaterial or the design reported in our paper.

• moreover, still on the physical aspects of the paper, the “metasurface” label seems off-topic in this work, besides pure “marketing”.

Response:

We appreciate the suggestion made by the reviewer and acknowledge his/her concern. However, we have no intention of marketing when using the term “metasurface.” The term “acoustic metasurfaces” have been used for over a decade, and *it is hardly something that we rely on for novelty.* However, the device we fabricate is a single layer of two-dimensional array of acoustic resonators with subwavelength thickness (3.5 cm, which roughly corresponds to $\lambda/5$ to $\lambda/10$). It satisfies the very definition of metasurface. Therefore, we do think that metasurface is the best term to describe the device. Hence with all due respect and after deliberations, we have chosen to retain the term in the paper. We sincerely hope that the reviewer could understand our choice. After careful consideration, we have revised our manuscript to accurately describe our resonator.

• the argument on “reverberant environment”: I doubt the practical example shown in this paper can extend beyond this relatively poorly reverberant environment: with the reverberation times reported in the Methods section ($T_{60} \sim 0.52$ s), I doubt that the tested room can be qualified as “reverberant”. It is especially obvious in Figures 3d, 4c,d,e and 6c,f, where the recorded signals show the dryness of the room...

Response:

We appreciate the reviewer’s comments. The reviewer mentioned that the room appears to be “dry”, which is due to the seemingly short T_{60} of $T_{60} \cong 0.52$ s. To clarify, our experimental setups do exhibit adequate reverberations.

First, the Schroeder frequency of the room is 217 Hz, meaning that the sound field is diffusive above this frequency.

Second, the reverberation time of our room is not short for the volume of the room. According to Sabine’s formula, smaller rooms typically have shorter reverberation times. For instance, as indicated in Ref. 5, the suggested data on reverberation time confirms that the reverberation time in our smaller-sized room is within a reasonable range. To facilitate reference, we have included Fig. R4, which replicates Figure 10.13 at page 538 in Ref. 5, with the placement of the reverberation time measurement in our laboratory labeled accordingly.

Fig. R4 The optimum reverberation time versus volume V (reproduced from Figure 10.13 at page 538 in Ref. 5).

Third, to fully address the reviewer's concerns, we have characterized the acoustics of the room. A reverberating acoustic environment is one that the sound intensity is homogeneous. To verify this property, we have conducted measurements of the sound fields in 8 different planes (1.5-by-1.4 m²) in the room in the frequency range of $f_1 = 250$ Hz to $f_2 = 8000$ Hz. The spectral average of sound pressure levels (SPL), defined as

$$SPL = 10 \log_{10} \left[\frac{1}{f_2 - f_1} \int_{f_1}^{f_2} |p(f)|^2 df \right], \quad (R2)$$

are plotted in Fig. R5. It is seen that the SPL is flat in space with some random undulations. To be specific, the spatially averaged SPL \overline{SPL} and standard deviation $\sigma(SPL)$ for the 1000-2000 Hz range are -28.96 dB and 0.6556 dB, respectively. For the 250-8000 Hz range, they are -25.86 dB and 0.5615 dB, respectively. It has been accepted that an acoustic field in a qualified reverberation room exhibits adequate diffuseness if the standard deviations remains under 1.5 dB⁶. Therefore, the experimental environment is a reverberating room. A reasonable explanation is that the multiple scattering in the room is sufficient, which can be expressed by the average number of scattering events, we define it as $N_s = \frac{\tau}{\Delta t_s} = \frac{cT_{60}}{6 \ln 10 \bar{\ell}}$

$\frac{343 \times 0.52}{6 \ln 10 \times 1.27} \approx 10$, where $\tau = T_{60}/(6 \ln 10)$, is the exponential decay time for which the sound energy in the room decays to $1/e$ of its initial value, $\Delta t_s = c/\bar{\ell}$ is the average time interval between two scattering events, c is the speed of sound, and $\bar{\ell}$ is the scattering mean free path. By utilizing our measured T_{60} , we can estimate N_s to be 10, which explains why the sound field we measured approximates a diffuse sound field. Even in cases where the reverberation time is short, multiple scattering can still be very significant if the mean free path is sufficiently small. This result agrees with the conclusion that sound field is random (presented in Methods and the Supplementary Note 2).

Fig. R5 The spatial distribution of the sound pressure levels (SPL) is illustrated for Planes 1-8. The calculations for the combined SPL are conducted for two frequency ranges: 1000-2000 Hz (upper) and 250-8000 Hz (bottom). Both frequency ranges exhibit a standard deviation of less than 0.7 dB.

Fourth, we think the possible reason for the signals presented in the paper to appear “dry” is due to the averaging of multiple independent experiment and due to the long pulse durations of the sound emitted by the sources ($>0.6s > T60$). To demonstrate, Fig. R6 shows the temporal signal received in the room from a short pulse with a duration of 0.1 s. A long echo is clearly observed.

Given the importance of this question to the reviewer, we have revised Supplementary Note 2 to provide detailed data to reflect the fact that the experimental room was well reverberant.

Fig. R6 The signal received by a microphone in the room for a source emitting a 0.1-s Gaussian pulse.

... Moreover, it would have been interesting to show how the performance degrade with reverberation time in the room.

Response:

Regarding this point, significant modifications to the reverberation time of the experimental setup in existing laboratories is difficult. Therefore, it is challenging for us to perform experimental study on this problem.

However, we can provide some analyses. Our method relies on a suitable level of reverberation. If there is no reverberation at all, the direct sound received by the microphone dominates, the reflective metasurfaces offers almost no control. For example, this experiment does not work in an anechoic chamber. On the other hand, excessively long reverberation time may also reduce performance. This is because very long reverberation times correspond to a greater number of reflections, leading to an increase in the correlation between the optimal states of two different resonators, resulting in a reduction in the number of controllable modes in the room, thereby reducing the performance of the reflector. Also, excessively long reverberation time implies that the Q-factor of the room is high such that the correlation bandwidth becomes narrow. As a result, for a single-frequency source, the number of modes excited decreases. Therefore, neither too low or too high level of reverberation is desirable.

The reviewer’s question is indeed interesting and thought-provoking. We intend to carry out relevant investigations in the future, when it is possible for us to modify the reverberation time of the room.

In response to this comment, we added a note to the Discussion section: *“Our method relies on moderate reverberation. Without reverberation, the direct sound received by the microphone dominates, rendering the ARMs almost ineffective. For instance, this experiment cannot be conducted in an anechoic chamber. On the contrary, excessively long reverberation time also affects performance by increasing the correlation between the optimal states of two different ARMs, resulting in a reduction in the number of controllable modes and consequently compromising the performance of the reflector. Therefore, both low and high levels of reverberation are undesirable.”*

- on the argument on broadbandness: 40 Hz at 1’300 Hz represents only 1/24th octave, which is far from a broad frequency range. The authors should be very prudent with such claims ...

Response:

Thank you for the critic. We agree with the reviewer and we have replaced “wideband / broadband” with “a finite-frequency band.” We have also presented a new result that show the effect achieve for a 100-

Hz band (Supplementary Fig. 8).

- I am not fully convinced by the demonstration, focusing on pure tones at discrete frequencies. This doesn't make a sound demonstration of application with respect to (speech) communication problems: what happens with real speech signals instead, or at least with band-limited modulated noises? This is not even mentioned in the paper, nor in the conclusions.

Response:

Thank you for the comment. Indeed, our paper reports a proof-of-principle experiment that acoustic communication be improved by controlling the complex sound field in reverberating space. This is a new concept that has not been demonstrated before. We do agree with the reviewer that the technology needs to be improved to handle real speech signals. Therefore, we have replaced the “dummy persons” in Figs. 1 and 6(a, d) with a loudspeaker to avoid overselling the results and eliminate any chance of confusion. Moreover, we modified the wording of the article to avoid exaggerating existing findings, such as replacing “speaker” with “loudspeaker” and “listener” with “microphone” in the text.

For this technology to further develop, the main challenge is clearly the bandwidth. To expand the controllable frequency range, one route is to further engineer the metasurfaces. In a simple way, we can fabricate addition units that control the reflected phases of sound at a different frequency regime. These metasurfaces can then work together with offer an increase controlling bandwidth.

To summarize, the work reported in this manuscript shows some interesting ideas and results, that performs accordingly to the theory but only for too limited acoustic signals, and in a rather “non-reverberant” situation. Then it fails to demonstrate an actual solution that could be representative of a real-life situation. Consequently and although the potential applicative outcomes could have been very interesting, it does not contain ground-breaking results and therefore it is not suited, in my opinion, for publication in a journal such as Nature Communications.

Response:

We appreciate the reviewer's comments and are glad that the results are interesting for the reviewer. We have shown that our room is indeed reverberant. It is important to highlight that this is the first instance in acoustic communication can be controlled and improved by modulating the sound field. While our current system showcases control over single and multi-frequencies, it is nevertheless a conceptual advancement over traditional means of room acoustics, and the first step towards reconfigurable of sound fields.

Reviewer #3 (Remarks to the Author):

This manuscript introduces a new approach to enhance multi-user sound communications in reverberant environments. The primary focus of this method is the mitigation of crosstalk between multiple talkers, facilitated by adaptable binary acoustic metasurfaces strategically positioned on room walls. The underlying challenge addressed in this study involves the interference arising from reverberation when multiple individuals speak simultaneously. Despite humans have the ability to selectively attend to specific speakers, known as the cocktail party effect, enhancing speech perception by minimizing crosstalk remains advantageous, which shows the scientific significance of this work.

Conventional strategies typically target crosstalk reduction at either the source (loudspeakers) or the receiver (microphones). For instance, cancellation filters are employed to attenuate crosstalk between loudspeakers in binaural or multi-channel audio setups, as discussed in A. Roginska, and P. Geluso (Eds.), *Immersive Sound: The Art and Science of Binaural and Multi-Channel Audio*, (London, Routledge, Taylor & Francis Group, 2018), and B. Xie, *Spatial Sound: Principles and Applications*, (Boca Raton, CRC Press, 2023). Signals captured by microphones are processed to extract distinct audio signals from each source, see in J. Benesty, M. M. Sondhi, and Y. Huang (Eds.), *Springer Handbook of Speech Processing*, (London, Springer, 2008). However, this study presents a new proposition: the elimination of crosstalk among transfer functions connecting sources and receivers. While the work has the potential to serve as a valuable source of inspiration for researchers, certain limitations need to be addressed, as outlined below:

Response:

We thank the reviewer for his/her careful review and objective description of our paper. We appreciate the reviewer for listing some interesting literature here. They have been added to the reference list of the revised paper and we added a note to the Discussion section:

“Our approach achieves channel isolation through the physical modulation of the reverberating sound field. This is unlike any traditional strategy that often relies on restricting the sources or the receivers^{29–31}, e.g., putting on a noise-blocking headsets. This research highlights that modifying the channel matrix during the crosstalk cancellation process can be an effective approach, offering new solutions and technological means in related fields.”

Below we do our best to answer his/her concerns and suggestions.

1. I suggest authors mention the cocktail party effect in the Introduction. For example, see the reference B. Arons, “A review of the cocktail party effect,” *Journal of the American Voice I/O society* 12(7), 35–50 (1992).

Response:

Thank you for the suggestion. We have cited this paper in the revised manuscript and we added a discussion in the introduction:

“The phenomenon of the cocktail party effect in the human auditory system allows individuals to selectively attend to specific sounds²³, facilitating the reduction of crosstalk in multi-channel communications. Nevertheless, in complex environments, the cognitive capacity of the human perception system is limited. ”

2. The current performance evaluation focuses on triple tones (Do Re Mi), which may not accurately reflect real-world scenarios where speech signals are often broadband (typically at least ranging from 500 Hz to

4 kHz). The effectiveness of the proposed system for wideband signals remains uncertain and merits further investigation/discussion.

Response:

Thank you for the comment. We are optimistic about the potential of using current technologies to real-world scenarios, although we acknowledge considerable challenges need to be overcome that requires significant advancement of the technology. In order to ensure accurate representation of our results, we have replaced the “dummy persons” depicted in Figs. 1 and 6(a, d) with a loudspeaker and we have replaced “speaker” with “loudspeaker” and “listener” with “microphone” in the text. This modification aims to avoid any overstatement of our findings and prevent any misconceptions.

To address this issue, we performed experiments to evaluate the potential of wideband control in existing systems. It is important to note that our study is still in the preliminary stage and unable to optimize channel isolation for the 500Hz to 4kHz bandwidth, which falls outside the operating frequency range of our current metasurfaces (1100-1850Hz). Nevertheless, we made attempts to optimize channel isolation for two limited bandwidth ranges: 1280Hz to 1320Hz with a bandwidth of 40Hz, and 1350Hz to 1450Hz with a bandwidth of 100Hz. Our previous Supplementary Information presented results for the 40-Hz bandwidth. For ease of comparison, we redraw it as Fig. R7a. We have further performed an experiment to demonstrate optimal 2-channel isolation for the 100-Hz bandwidth, as shown in Fig. R7b. In this experiment, The idea is to simultaneously optimize channel matrices at 26 consecutive frequencies with an interval of 4 Hz, such that a 100-Hz continuous band is controlled. The outcome clearly shows that this “brute force” approach is still effective, as indicated by the improvements in diagonalization and effective rank. But the overall effect has already degraded. We therefore think that this could be the upper limit of our current technology in terms of bandwidth.

We think at least two routes can lead to further improvement of the bandwidth. First, because the metasurfaces occupy only less than 10% of the room’s surface area, it is possible to further increase the working bandwidth by incorporating additional metasurfaces that control the sound field at different frequencies. Second, increasing the phase-modulation steps of each unit cell (right now we use only binary phase modulation), which effectively offers more refined phase control, may improve the outcome.

Fig. R7 Optimization of 2×2 channel matrices by minimizing $G_1(H)$ over a continuous band of frequency spanning 40 Hz (a) and 100 Hz (b).

3. The optimization of crosstalk cancellation is centered around specific frequencies, leading to impressive results in some instances (e.g., Fig. 3a). However, it is likely that this targeted optimization could negatively impact performance in other frequency ranges, possibly adversely enhance crosstalk in those frequency ranges. To provide a comprehensive understanding, it is recommended that the authors present channel isolation metrics across a wider frequency spectrum (500 Hz to 4 kHz at least).

Response:

The issue raised by the reviewer is indeed important. From a statistical perspective, the properties will converge to the Rayleigh channel characteristics, which means the change induced in other frequency ranges is unintentionally random, and thus the combined outcomes do not have significant impacts on the channel isolation metrics. By the reviewer’s suggestion, we plot the channel isolation metrics across in 500-4000Hz in Fig. R8. It is clearly that only the targeted frequency of 1300 Hz is affected. This new figure is also included in the revised Supplementary Information as Supplementary Fig. 7.

Fig. R8 Measurement of R_{eff} and w_1 in the frequency range of 500-4000 Hz, with optimization performed solely at 1300 Hz.

4. In audio engineering, presenting results in decibels (dB) instead of a linear scale (e.g., Fig. 4(c-e)) would offer a more meaningful representation, aiding readers in grasping the magnitude of improvements with greater clarity.

Response:

We appreciate the suggestion from the reviewer. However, after some deliberation, we have decided to retain the linear scale. The reason is that channel isolation not only suppresses the signal from the unwanted source but also boosts the signal from the desirable source. The decibel scale is optimal for showing the suppression because it emphasizes small data, but it is not ideal for showing the improvement.

To better show this, we have additionally plotted the data in the main text in decibel scale in Fig. R9-12. It is quite clear that the suppression can be very easily identified but the improvements appear to be

almost invisible. For this reason, we have retained the linear scale. To comply with the reviewer's request, we have marked the corresponding decibel values in the Figs. 3, 4 and 6 in the revised manuscript.

Fig. R9 Time-domain envelope of signals (Fig. 3d in the main text) captured by microphone 1 (a) and microphone 2 (b) displayed in decibels.

Fig. R10 Time-domain envelope of signals (Fig. 4d-e) captured by microphone 1 (a, c) and microphone 2 (b, d) displayed in decibels.

Fig. R11 Time-domain envelope of signals (Fig. 6a-c) captured by microphone 1-6 displayed in decibels.

Fig. R12 Time-domain envelope of signals (Fig. 6d-f) captured by microphone 1-6 displayed in decibels.

Reference

1. Yon, S., Tanter, M. & Fink, M. Sound focusing in rooms: The time-reversal approach. *The Journal of the Acoustical Society of America* **113**, 1533–1543 (2003).
2. Ma, G. *et al.* Towards anti-causal Green's function for three-dimensional sub-diffraction focusing. *Nature Phys* **14**, 608–612 (2018).
3. Ma, G., Fan, X., Sheng, P. & Fink, M. Shaping reverberating sound fields with an actively tunable metasurface. *Proceedings of the National Academy of Sciences* **115**, 6638–6643 (2018).
4. Wang, Q., del Hougne, P. & Ma, G. Controlling the spatiotemporal response of transient reverberating sound. *Phys Rev Appl* **17**, 044007 (2022).
5. Beranek, L. L. & Mellow, T. J. *Acoustics: sound fields, transducers and vibration*. (Academic Press, 2019).
6. Nélisse, H. & Nicolas, J. Characterization of a diffuse field in a reverberant room. *The Journal of the Acoustical Society of America* **101**, 3517–3524 (1997).

Reviewer #1 (Remarks to the Author):

I have spent some time in going over the response by the corresponding author to my comments. I am impressed by the care and the extra information provided in the response to each of my three questions. I am satisfied by the response, and recommend the manuscript for publication.

I have also looked over the response to the comments of the other two referees. They are equally convincing to me.

Reviewer #2 (Remarks to the Author):

The authors have provided detailed point-by-point answers to my concerns, most of which are convincing, except maybe the reference to a "reverberant" sound field and the "metasurface" label, which I further argue in my following comments. I would suggest that the authors employ the term "diffuse" or "diffuseness" instead, as I believe it is more appropriate here, and that they take some precaution while using the term "metasurface".

Argumentation 1: To begin, we would like to argue that the cost functions are clearly not ad hoc. The formulation of the cost functions rests on the systematic analysis of channel matrix, whose foundation is the theories of information and communications. Our formulation is general in that it can be applied to any reverberating environment and it can be used to handle any number of sources and receivers placed anywhere in the environment. Indeed, what we demonstrate here is not the general control of global acoustic properties but the refined control of the local sound fields for specific purposes, so the changes one has made is to account for the diverse communication scenarios. It clear does not mean the approach is ad hoc.

==> My reaction: I acknowledge the term "ad hoc" was not suited and accept the argumentation.

Argumentation 2: About the novelty of our work. We use information and communication theories to guide the active control of reverberating sound fields. This conceptually new approach has not been adapted for acoustics previously, to the best of our knowledge. In addition, we present the first demonstration of acoustic wavefield shaping for multi-frequency signals using passive reflective units, which has not been demonstrated before either.

==> My reaction: I acknowledge the approach is sufficiently novel and accept this argumentation.

Argumentation 3: First, the Helmholtz resonators we proposed should not be considered as active diffusers. Diffusers should ideally scatter waves evenly to all directions, such that a homogenous wavefield can be obtained. However, the functionality of our tunable metasurface is completely different: its goal is to scatter waves in specific ways such that the reverberating wavefield, which is already homogeneous, is purposely disrupted so the outcome is more suitable for multi-channel communications. In fact, in some particular realizations, it is possible for a good number of units to end up in the same state (Fig. R3), so the corresponding portion of metasurface produces near specular reflection, which is clearly different from any diffuser. In response to this comment, we added a note to the Discussion section:

"We also remark that the ARMs should not be considered as diffusers. Its function is not to scatter waves evenly in all directions for the formation of a uniform wave field. Instead, it scatters waves in specific ways designed to intentionally disrupt an already uniform reflected wave field, thereby achieving OCI."

Second, we would like to clarify that our metasurface consisting of Helmholtz resonators was designed and constructed in house specifically for this experiment, instead of utilizing any readily

available assets in our lab. In fact, this is the first time that our group works on Helmholtz resonators. And the Helmholtz resonator is distinct in its tunable design from those reported in existing literature. In our previous works about wavefield shaping in reverberating rooms, the modulation was implemented by membrane-type acoustic metamaterials, which only offers modulation the transmitted sounds in <1 kHz regime. Hence, they must be placed somewhere in the middle of the room. The device used in this work, however, modulates the reflected waves, so they can be regarded as tunable boundaries of the room.

Third, the music demonstration and the 40-Hz continuous band experiment (replaced by 100-Hz bandwidth experiment in revised version) already show that the capability of our two-phase-controlling reflector is not limited to two specific operating frequencies. The reflection phase can be actively tuned across a wide frequency range of 1100-1850 Hz, which is over 2/3 of an octave. This is shown by the black dashed curve in the upper panel of Fig. 2d in the main text.

Lastly, we are not aware of the "other solutions" mentioned by the reviewer. To the best of our knowledge, acoustic wavefield shaping in reverberating cavity has only been achieved using either the membrane-type metamaterial or the design reported in our paper.

=> My reaction: It is true that the proposed ARM design is different from the membrane resonators employed in their former work, and I acknowledge that this concept has been specifically designed for the purpose of this experiment. However, I still think there are alternative strategies, employing active membrane-based unit-cells, that could allow much more versatile tunability, reason why I am a bit dubious on the timeliness of this solution. I especially think of the concept of active acoustic resonators that allows controlling acoustic impedances in amplitude and phase over an extended bandwidth (sometimes exceeding multiple octaves). However, I agree that the proposed solution is a simple step towards achieving the targeted manipulation of sound fields, and accept their argumentation. I would maybe ask the authors have a look on the existing literature on reconfigurable/active acoustic resonators (for example in the non-exhaustive list below).

PS: please note that the [1'100 - 1'850 Hz] interval corresponds to 3/4 of an octave, not 2/3...

[1] Popa, B.I., Shinde, D., Konneker, A., Cummer S.A. Active acoustic metamaterials reconfigurable in real time. *Phys. Rev. B* 91, 220303(R) (2015).

<https://link.aps.org/doi/10.1103/PhysRevB.91.220303>

[2] Lissek, H., Rivet, E., Laurence, T., Fleury, R. Toward wideband steerable acoustic metasurfaces with arrays of active electroacoustic resonators. *J. Appl. Phys.* 123 (9): 091714 (2018).

<https://doi.org/10.1063/1.5011380>

[3] Koutserimpas, T., Rivet, E., Lissek, H., Fleury, R. Active Acoustic Resonators with Reconfigurable Resonance Frequency, Absorption, and Bandwidth. *Phys. Rev. Applied* 12, 054064 – P(2019). <https://link.aps.org/doi/10.1103/PhysRevApplied.12.054064>

[4] Tang, X., Liang, S., Jiang, Y. et al. Magnetoactive acoustic metamaterials based on nanoparticle-enhanced diaphragm. *Sci Rep* 11, 22162 (2021). <https://doi.org/10.1038/s41598-021-01569-9>

Argumentation 4: We appreciate the suggestion made by the reviewer and acknowledge his/her concern. However, we have no intention of marketing when using the term "metasurface." The term "acoustic metasurfaces" have been used for over a decade, and it is hardly something that we rely on for novelty. However, the device we fabricate is a single layer of two-dimensional array of acoustic resonators with subwavelength thickness (3.5 cm, which roughly corresponds to $\lambda/5$ to $\lambda/10$). It satisfies the very definition of metasurface. Therefore, we do think that metasurface is the best term to describe the device. Hence with all due respect and after deliberations, we have chosen to retain the term in the paper. We sincerely hope that the reviewer could understand our choice. After careful consideration, we have revised our manuscript to accurately describe our resonator.

=> My reaction: Sorry but I did not suspect the authors of marketing, but it was rather a general observation that the term "metasurface" was often subject to abusive use, leading to discredit the term. Also the definition of a metasurface is not only that it has "subwavelength" dimension, but

also that it is designed so as to formally bend the laws of physics (eg. Snell-Descartes law of refraction). Here the reference to a specific physical mechanism is absent from the discussion, but only through the "communication theory framework" (which resembles a black-box in terms of physical insight), reason why I argued the term "metasurface" may be seen as abusive. If the authors don't find another appropriate term for designating the device, I would suggest they take some precaution in introducing the term.

Argumentation 5: We appreciate the reviewer's comments. The reviewer mentioned that the room appears to be "dry", which is due to the seemingly short T60 of $T60 \approx 0.52$ s. To clarify, our experimental setups do exhibit adequate reverberations.

First, the Schroeder frequency of the room is 217 Hz, meaning that the sound field is diffusive above this frequency.

Second, the reverberation time of our room is not short for the volume of the room. According to Sabine's formula, smaller rooms typically have shorter reverberation times. For instance, as indicated in Ref. 5, the suggested data on reverberation time confirms that the reverberation time in our smaller-sized room is within a reasonable range. To facilitate reference, we have included Fig. R4, which replicates Figure 10.13 at page 538 in Ref. 5, with the placement of the reverberation time measurement in our laboratory labeled accordingly.

Third, to fully address the reviewer's concerns, we have characterized the acoustics of the room. A reverberating acoustic environment is one that the sound intensity is homogeneous. To verify this property, we have conducted measurements of the sound fields in 8 different planes (1.5-by-1.4 m²) in the room in the frequency range of $f_1=250$ Hz to $f_2=8000$ Hz. The spectral average of sound pressure levels (SPL), defined as $SPL=10\log_{10}[1/f_2-f_1 \int |p(f)|^2 df]$, (R2) are plotted in Fig. R5. It is seen that the SPL is flat in space with some random undulations. To be specific, the spatially averaged SPL \overline{SPL} and standard deviation $\sigma(SPL)$ for the 1000-2000 Hz range are -28.96 dB and 0.6556 dB, respectively. For the 250-8000 Hz range, they are -25.86 dB and 0.5615 dB, respectively. It has been accepted that an acoustic field in a qualified reverberation room exhibits adequate diffuseness if the standard deviations remains under 1.5 dB⁶. Therefore, the experimental environment is a reverberating room. A reasonable explanation is that the multiple scattering in the room is sufficient, which can be expressed by the average number of scattering events, we define it as $N_s = \tau \Delta t_s = c T60 / 6 \ln 10$
 $\approx 343 \times 0.526 \ln 10 \times 1.27 \approx 10$, where $\tau = T60 / (6 \ln 10)$, is the exponential decay time for which the sound energy in the room decays to $1/e$ of its initial value, $\Delta t_s = c / \bar{v}$ is the average time interval between two scattering events, c is the speed of sound, and \bar{v} is the scattering mean free path. By utilizing our measured T60, we can estimate N_s to be 10, which explains why the sound field we measured approximates a diffuse sound field. Even in cases where the reverberation time is short, multiple scattering can still be very significant if the mean free path is sufficiently small. This result agrees with the conclusion that sound field is random (presented in Methods and the Supplementary Note 2).

Fourth, we think the possible reason for the signals presented in the paper to appear "dry" is due to the averaging of multiple independent experiment and due to the long pulse durations of the sound emitted by the sources ($>0.6s > T60$). To demonstrate, Fig. R6 shows the temporal signal received in the room from a short pulse with a duration of 0.1 s. A long echo is clearly observed. Given the importance of this question to the reviewer, we have revised Supplementary Note 2 to provide detailed data to reflect the fact that the experimental room was well reverberant.

==> My reaction: I think the authors make a confusion between "reverberant" and "diffuse". For instance, a "naked" reverberant chamber, although designed for it, might not be "diffuse" enough, reason why diffusers are actually used to adapt the "diffuseness" of the room to standards. Also, the argument after which the reverberation time of the room exceeds the "optimal" rt for its volume is not acceptable (a "dry" classroom or recording studio may have an "optimal" reverberation for its use, but it won't be qualified "reverberant"). To precise my thoughts, I fear that the term "reverberant" is misleading, and may be a source of mockery to actual acoustic specialists. My recommendation would be to prefer the term "diffuse" (which is the actual characteristic satisfied by Fig. R5 by the way) instead of "reverberant".

Argumentation 6: Regarding this point, significant modifications to the reverberation time of the experimental setup in existing laboratories is difficult. Therefore, it is challenging for us to perform experimental study on this problem.

However, we can provide some analyses. Our method relies on a suitable level of reverberation. If there is no reverberation at all, the direct sound received by the microphone dominates, the reflective metasurfaces offers almost no control. For example, this experiment does not work in an anechoic chamber. On the other hand, excessively long reverberation time may also reduce performance. This is because very long reverberation times correspond to a greater number of reflections, leading to an increase in the correlation between the optimal states of two different resonators, resulting in a reduction in the number of controllable modes in the room, thereby reducing the performance of the reflector. Also, excessively long reverberation time implies that the Q-factor of the room is high such that the correlation bandwidth becomes narrow. As a result, for a single-frequency source, the number of modes excited decreases. Therefore, neither too low or too high level of reverberation is desirable.

The reviewer's question is indeed interesting and thought-provoking. We intend to carry out relevant investigations in the future, when it is possible for us to modify the reverberation time of the room.

In response to this comment, we added a note to the Discussion section: "Our method relies on moderate reverberation. Without reverberation, the direct sound received by the microphone dominates, rendering the ARMs almost ineffective. For instance, this experiment cannot be conducted in an anechoic chamber. On the contrary, excessively long reverberation time also affects performance by increasing the correlation between the optimal states of two different ARMs, resulting in a reduction in the number of controllable modes and consequently compromising the performance of the reflector. Therefore, both low and high levels of reverberation are undesirable."

==> My reaction: I acknowledge this parametric study is not easily accessible to experiments, but a model would have been a good support for it. I accept the argumentation.

Argumentation 7: Thank you for the critic. We agree with the reviewer and we have replaced "wideband / broadband" with "a finite-frequency band." We have also presented a new result that show the effect achieve for a 100 Hz band (Supplementary Fig. 8).

==> My reaction: thank you.

Argumentation 8: Thank you for the comment. Indeed, our paper reports a proof-of-principle experiment that acoustic communication be improved by controlling the complex sound field in reverberating space. This is a new concept that has not been demonstrated before. We do agree with the reviewer that the technology needs to be improved to handle real speech signals. Therefore, we have replaced the "dummy persons" in Figs. 1 and 6(a, d) with a loudspeaker to avoid overselling the results and eliminate any chance of confusion. Moreover, we modified the wording of the article to avoid exaggerating existing findings, such as replacing "speaker" with "loudspeaker" and "listener" with "microphone" in the text.

For this technology to further develop, the main challenge is clearly the bandwidth. To expand the controllable frequency range, one route is to further engineer the metasurfaces. In a simple way, we can fabricate addition units that control the reflected phases of sound at a different frequency regime. These metasurfaces can then work together with offer an increase controlling bandwidth.

==> My reaction: I accept the argumentation.

Argumentation 9: We appreciate the reviewer's comments and are glad that the results are interesting for the reviewer. We have shown that our room is indeed reverberant. It is important to

highlight that this is the first instance in acoustic communication can be controlled and improved by modulating the sound field. While our current system showcases control over single and multi-frequencies, it is nevertheless a conceptual advancement over traditional means of room acoustics, and the first step towards reconfigurable of sound fields.

==> My reaction: I accept the argumentation, and would be favourable to the publication of a revised manuscript, provided the authors take into consideration my remaining concerns on the denomination of "reverberant" sound fields, and a proper justification of their use of the "metasurface" label.

Reviewer #3 (Remarks to the Author):

The authors have addressed most of my comments and therefore I recommend this manuscript for publication in NC.

However, I want to point out one thing for the authors to consider, regarding my previous 2nd comment. The proposed method's mechanism aims to mitigate crosstalk in the transfer function between sources and receivers. This involves utilizing the reflection path from the wall to cancel the direct sound path from the source to the unwanted receiver. The metasurface on the wall is able to modify the reflection path but cannot influence the direct sound from the source to the unwanted receiver. While this has minimal adverse effects on pure-tone or narrow-band speech signals, it is likely to have significant adverse impacts on broadband speech signals.

To illustrate, consider that the direct sound always arrives at the unwanted receiver before the reflected sound due to the shorter path. Here, causality imposes an upper limit on the crosstalk cancellation achievable by the proposed method. I would recommend that the authors acknowledge this potential drawback in the discussion.

Summary of major revisions:

Revisions	Location	Addressing
Replaced the term “reverberating environment” to “indoor environment”, “reverberating field” to “complex field”, and “reverberating room” to “room”.	Main text	Reviewer 2
A discussion on active strategy to improve the performance of unitcells has been included. Relevant references are included.	Main text, discussion (Page 8), Ref. [34-37].	Reviewer 2
Added the discussion on the direct sound on the performance.	Main text, discussion (Page 8).	Reviewer 3

Reviewer #1

I have spent some time in going over the response by the corresponding author to my comments. I am impressed by the care and the extra information provided in the response to each of my three questions. I am satisfied by the response, and recommend the manuscript for publication.

I have also looked over the response to the comments of the other two referees. They are equally convincing to me.

Response:

We thank the reviewer for the effort in assessing our paper and for the positive recommendation.

Reviewer #2

The authors have provided detailed point-by-point answers to my concerns, most of which are convincing, except maybe the reference to a “reverberant” sound field and the “metasurface” label, which I further argue in my following comments. I would suggest that the authors employ the term “diffuse” or “diffuseness” instead, as I believe it is more appropriate here, and that they take some precaution while using the term “metasurface”.

Response:

We are most grateful to the reviewer for his/her careful evaluation. Below, we present point-to-point answers to the comments. For the brevity of this letter, we keep only the reviewer’s reactions but not the arguments from our own previous response.

Argumentation 1: ...

==> My reaction: I acknowledge the term “ad hoc” was not suited and accept the argumentation.

Response:

Thank you.

Argumentation 2: ...

==> My reaction: I acknowledge the approach is sufficiently novel and accept this argumentation.

Response:

We thank the reviewer for his/her acceptance of this perspective.

Argumentation 3: ...

==> My reaction: It is true that the proposed ARM design is different from the membrane resonators employed in their former work, and I acknowledge that this concept has been specifically designed for the purpose of this experiment. However, I still think there are alternative strategies, employing active membrane-based unit-cells, that could allow much more versatile tunability, reason why I am a bit dubious on the timeliness of this solution. I especially think of the concept of active acoustic resonators that allows controlling acoustic impedances in amplitude and phase over an extended bandwidth (sometimes exceeding multiple octaves). However, I agree that the proposed solution is a simple step towards

achieving the targeted manipulation of sound fields, and accept their argumentation. I would maybe ask the authors have a look on the existing literature on reconfigurable/active acoustic resonators (for example in the non-exhaustive list below).

PS: please note that the [1'100 - 1'850 Hz] interval corresponds to 3/4 of an octave, not 2/3...

[1] Popa, B.I., Shinde, D., Konneker, A., Cummer S.A. Active acoustic metamaterials reconfigurable in real time. Phys. Rev. B 91, 220303(R) (2015). <https://link.aps.org/doi/10.1103/PhysRevB.91.220303>

[2] Lissek, H., Rivet, E., Laurence, T., Fleury, R. Toward wideband steerable acoustic metasurfaces with arrays of active electroacoustic resonators. J. Appl. Phys. 123 (9): 091714 (2018). <https://doi.org/10.1063/1.5011380>

[3] Koutserimpas, T., Rivet, E., Lissek, H., Fleury, R. Active Acoustic Resonators with Reconfigurable Resonance Frequency, Absorption, and Bandwidth. Phys. Rev. Applied 12, 054064 – P(2019). <https://link.aps.org/doi/10.1103/PhysRevApplied.12.054064>

[4] Tang, X., Liang, S., Jiang, Y. et al. Magnetoactive acoustic metamaterials based on nanoparticle-enhanced diaphragm. Sci Rep 11, 22162 (2021). <https://doi.org/10.1038/s41598-021-01569-9>

Response:

We thank the reviewer for his/her comments. We acknowledge that future applications of active approaches could be a potential improvement direction. In response to this suggestion, we have included additional discussions in the main text and incorporated the newly recommended references into the citation list:

“Finally, other active acoustic designs are potentially suitable for achieving similar functionalities in sound-field manipulations³⁴⁻³⁷.”

Furthermore, we would like to thank the reviewer for pointing out the error in the bandwidth calculation. It has been corrected in the main text.

Argumentation 4: ...

==> My reaction: Sorry but I did not suspect the authors of marketing, but it was rather a general observation that the term “metasurface” was often subject to abusive use, leading to discredit the term. Also the definition of a metasurface is not only that it has “subwavelength” dimension, but also that it is designed so as to formally bend the laws of physics (eg. Snell-Descartes law of refraction). Here the reference to a specific physical mechanism is absent from the discussion, but only through the “communication theory framework” (which resembles a black-box in terms of physical insight), reason why I argued the term “metasurface” may be seen as abusive. If the authors don't find another appropriate term for designating the device, I would suggest they take some precaution in introducing the term.

Response:

Thank you for your feedback. The defining criterion for the term “metasurface” lies in whether these structures possess particular wave-controlling functionalities that go beyond those of ordinary boundaries or surfaces. And it is generally accepted that such functionalities shall originate from the response from subwavelength building blocks. As such, metasurfaces encompass ever-expanding functionalities, ranging from the early demonstration of anomalous reflection^{1,2} and refraction^{3,4}, total absorption⁵, to hologram⁶ and more sophisticated wavefront shaping⁷. In fact, our own previous work in optimization-based wavefield shaping in a room was also achieved using transmission-type metasurfaces⁸, and for the five-year time since its publication, there has not been a single negative comment on the usage of the term “metasurface.” Therefore, we believe that the term “metasurface” is the most suitable term to our design. We do hope the reviewer can recognize that “metasurface” is an ever-evolving concept.

Argumentation 5: ...

==> My reaction: I think the authors make a confusion between “reverberant” and “diffuse”. For instance, a “naked” reverberant chamber, although designed for it, might not be “diffuse” enough, reason why diffusers are actually used to adapt the “diffuseness” of the room to standards. Also, the argument after which the reverberation time of the room exceeds the “optimal” rt for its volume is not acceptable (a “dry” classroom or recording studio may have an “optimal” reverberation for its use, but it won’t be qualified “reverberant”). To precise my thoughts, I fear that the term “reverberant” is misleading, and may be a source of mockery to actual acoustic specialists. My recommendation would be to prefer the term “diffuse” (which is the actual characteristic satisfied by Fig. R5 by the way) instead of “reverberant”.

Response:

Thank you for the comments. In response to the reviewer’s comments, we have modified relevant terms, such as changing the term “reverberating environment” to “indoor environment”, “reverberating field” to “complex field”, and “reverberating room” to “room”.

Also, thank you for pointing out the matter of diffuseness. Although diffuseness is indeed an important benchmark for acoustic environment, using this term in our paper may generate confusion. This is because, in the context of wave propagation in complex media, diffusion wave is described by the diffusion equation, which is a partial differential equation that is second order in space but first order in time^{9,10}. This clearly does not capture the characteristics of the sound fields in the room: the sound fields are harmonic in both space and time. Hence, we refrain from using the term “diffuse” or “diffusion.”

Argumentation 6: ...

==> My reaction: I acknowledge this parametric study is not easily accessible to experiments, but a model would have been a good support for it. I accept the argumentation.

Response:

We thank the reviewer for his/her acceptance of this perspective.

Argumentation 7: ...

==> My reaction: thank you.

Response:

We thank the reviewer for accepting this revision.

Argumentation 8: ...

==> My reaction: I accept the argumentation.

Response:

Thank you.

Argumentation 9: ...

==> My reaction: I accept the argumentation, and would be favourable to the publication of a revised manuscript, provided the authors take into consideration my remaining concerns on the denomination of “reverberant” sound fields, and a proper justification of their use of the “metasurface” label.

Response:

Thank you for recommending the acceptance of our paper. The two additional concerns have been addressed, and revisions have been made accordingly.

Reviewer #3

The authors have addressed most of my comments and therefore I recommend this manuscript for publication in NC.

Response:

We thank the reviewer for his/her careful review and publication suggestions.

However, I want to point out one thing for the authors to consider, regarding my previous 2nd comment. The proposed method's mechanism aims to mitigate crosstalk in the transfer function between sources and receivers. This involves utilizing the reflection path from the wall to cancel the direct sound path from the source to the unwanted receiver. The metasurface on the wall is able to modify the reflection path but cannot influence the direct sound from the source to the unwanted receiver. While this has minimal adverse effects on pure-tone or narrow-band speech signals, it is likely to have significant adverse impacts on broadband speech signals.

To illustrate, consider that the direct sound always arrives at the unwanted receiver before the reflected sound due to the shorter path. Here, causality imposes an upper limit on the crosstalk cancellation achievable by the proposed method. I would recommend that the authors acknowledge this potential drawback in the discussion.

Response:

We appreciate the reviewer for his/her valuable suggestion. Excessively strong direct sound may indeed compromise the effectiveness of our method. This is why all experiments were conducted with the speakers and microphones separated by a distance beyond the reverberation radius. At locations beyond the reverberation radius from a source, direct sound is at a similar or a weaker level compared to multiply scattered sound. This information is provided in the Methods section since the first submission. In light of the reviewer's suggestion, we do agree that it should be discussed in the main text as well. Therefore, we have added a note to the Discussion section of the manuscript to acknowledge this limitation:

"Therefore, for optimal performance, the sources and the receivers shall be greater than the reverberation radius (0.5 m in the existing experimental configuration)."

References

1. Li, Y., Liang, B., Gu, Z., Zou, X. & Cheng, J. Reflected wavefront manipulation based on ultrathin planar acoustic metasurfaces. *Scientific Reports* **3**, (2013).
2. Li, Y. *et al.* Experimental Realization of Full Control of Reflected Waves with Subwavelength

- Acoustic Metasurfaces. *Physical Review Applied* **2**, (2014).
3. Li, Y. *et al.* Tunable Asymmetric Transmission via Lossy Acoustic Metasurfaces. *Phys. Rev. Lett.* **119**, 035501 (2017).
 4. Yu, N. *et al.* Light Propagation with Phase Discontinuities: Generalized Laws of Reflection and Refraction. *Science* **334**, 333–337 (2011).
 5. Ma, G., Yang, M., Xiao, S., Yang, Z. & Sheng, P. Acoustic metasurface with hybrid resonances. *Nature Materials* **13**, 873–878 (2014).
 6. Xie, Y. *et al.* Acoustic Holographic Rendering with Two-dimensional Metamaterial-based Passive Phased Array. *Scientific Reports* **6**, (2016).
 7. Jang, M. *et al.* Wavefront shaping with disorder-engineered metasurfaces. *Nature Photonics* **12**, 84–90 (2018).
 8. Ma, G., Fan, X., Sheng, P. & Fink, M. Shaping reverberating sound fields with an actively tunable metasurface. *Proceedings of the National Academy of Sciences* **115**, 6638–6643 (2018).
 9. Mandelis, A. Diffusion Waves and their Uses. *Physics Today* **53**, 29–34 (2000).
 10. Mandelis, A. *Diffusion-Wave Fields*. (Springer New York, 2001).